# Scalable RF Simulation in Generative 4D Worlds

**Zhiwei Zheng** [1]  **Dongyin Hu** [1]  **Mingmin Zhao** [1]

## Abstract

Radio Frequency (RF) sensing has emerged as a powerful, privacy-preserving alternative to vision-based methods for various perception tasks. However, building high-quality RF datasets in dynamic and diverse environments remains a major challenge. To address this, we introduce WAVEVERSE, a prompt-based, scalable framework that simulates realistic RF signals from generated indoor scenes with human motions guided by spatial paths, enabling diverse and feasible behaviors without manual trajectory design. WAVEVERSE features a language-guided 4D world generator and a physics-based signal simulator that enables realistic simulation of RF signals in diverse environments. It employs a phase-coherent ray tracer that preserves both spatial and temporal phase consistency. The simulated signals show high fidelity on phase-sensitive benchmarks, and closely align with both real-world collected measurements and simulations from a proprietary electromagnetic solver. When used for data augmentation, WAVEVERSE consistently improves performance in downstream tasks like RF imaging and human activity recognition, with gains that grow with the amount of simulated data and surpass existing methods. Code and additional materials are available on the webpage.

## 1. Introduction

Radio Frequency (RF) sensing has emerged as a compelling modality for multiple tasks such as 3D imaging, human activity recognition, and health monitoring (Singh et al., 2019; Zhao et al., 2021; Lai et al., 2024). In safety-critical or low-visibility scenarios, RF-based methods (Sun et al., 2021a;b; Lai et al., 2025) still offer reliable imaging performance despite fog, smoke, or occlusion. At the same time, RF sensors do not capture images or videos, making

them inherently privacy-preserving and well-suited for contactless and continuous health monitoring, including vital sign monitoring (Zhao et al., 2016; Ha et al., 2020), sleep analysis (Zhao et al., 2017; He et al., 2025), and mental health assessment (Ha et al., 2021; Liang et al., 2023; Hu et al., 2025). Despite these advantages, acquiring large-scale and high-quality RF sensing datasets remains challenging. Building such datasets requires capturing a wide range of room layouts, human activities, and individual differences, all of which demand high cost and effort. Worse still, RF sensing systems differ widely in hardware configurations (i.e., bandwidth, antenna layout, and signal modulation), making it difficult to reuse data across systems. As a result, unlike vision or audio, RF sensing lacks standardized and unified benchmarks, limiting cross-system generalization and slowing research progress.

Recent efforts have explored both physics-based simulation (Cai et al., 2020; Zhang et al., 2022) and learning-based synthesis (Chen & Zhang, 2023; Chi et al., 2024) to address the challenges. However, early approaches focus on signal interactions with human bodies while neglecting the surrounding environment. This is problematic for RF sensing, where multipath propagation (i.e., multi-bounce reflections with surrounding structures like walls, floors, and objects) greatly affects the received signal and is a key factor limiting generalization (Wang et al., 2020; Zhang et al., 2023a). Moreover, learning-based synthesis (Chen & Zhang, 2023; Chi et al., 2024) typically requires a large training dataset to begin with and does not readily generalize beyond a specific predefined sensor configuration.

In this paper, we introduce WAVEVERSE, a hybrid generation–simulation framework to synthesize realistic and diverse RF signals. As illustrated in Fig. 1, WAVEVERSE combines 4D world generation with physics-based RF simulation. Specifically, it leverages the emergent capabilities of Large Language Models (LLMs) (Achiam et al., 2023; Hurst et al., 2024), shown in various tasks (Yang et al., 2024; Sushil et al., 2024), to generate diverse 3D indoor environments and introduce a scalable framework for synthesizing dynamic human motions within them. Given the 4D world (i.e., a 3D environment with dynamic human motions), WAVEVERSE employs a customized ray tracing engine that models multipath propagation and provides phase-accurate signals across antennas and over time. This

[1]University of Pennsylvania. Correspondence to: Mingmin Zhao <mingminz@cis.upenn.edu>.

*Proceedings of the 43rd International Conference on Machine Learning*, Seoul, South Korea. PMLR 306, 2026. Copyright 2026 by the author(s).

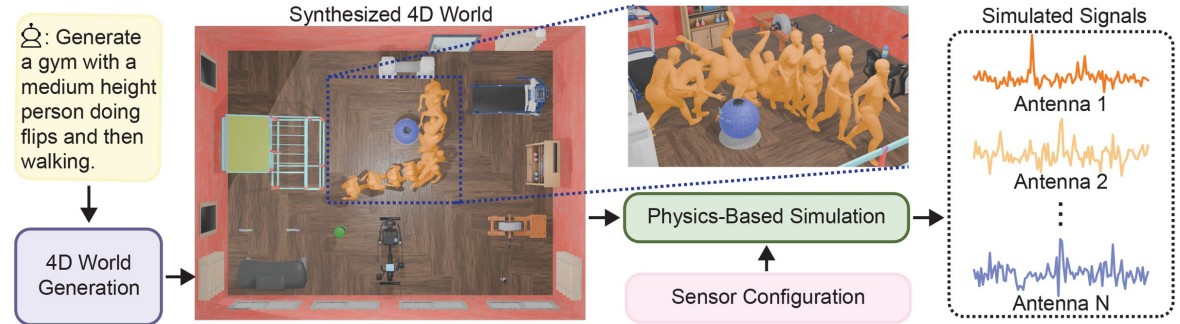

*Figure 1.* Given input text and sensor configuration, WAVEVERSE generates 4D worlds with moving humans in 3D environments and simulates the received RF signals using physics-based modeling.

hybrid design combines the best of both worlds: generative diversity from 4D synthesis and physical realism from RF simulation. The use of explicit mesh representations for 3D layouts provides additional benefits. It enables tightly aligned supervision for RF learning tasks (e.g., depth estimation, semantic segmentation, human poses) and supports RF simulation with flexible sensor configurations for a wide range of downstream applications, which are difficult to realize with existing methods.

WAVEVERSE introduces two key innovations to achieve these new capabilities. The first is spatial conditioning for human motion generation. Prior approaches (Xie et al., 2024; Dai et al., 2024) condition motion generation on *trajectories*, which are *time-indexed* sequences of joint positions. These trajectories prescribe not only *where* a person moves but also *when and how fast*, encoding velocities, durations, and frame-level details. While effective for strict and fine-grained control, this formulation is over-constrained, requires substantial manual effort to design, and ultimately restricts generative models from producing diverse and natural motions conditioned on the trajectories. In contrast, we introduce a *path*-based conditioning strategy that provides spatial guidance *without temporal assignment*. A path is defined as a set of waypoints specifying *where* the motion should occur, while leaving velocity, style, and duration flexible. This simpler representation enables automatic path generation and eliminates the need for manual trajectory design. We adopt a state-aware causal transformer that conditions each motion token prediction on the current spatial state for path-conditioned motion generation. Compared to baselines, it achieves the lowest path error and ending error, which measure the deviation over the full path following and the position difference at the final timestamp, and conforms well to the environment, as discussed in § 4.1.

Our second innovation is a physics-based simulation framework with phase-coherent ray tracing, which enables accurate and consistent modeling of signal phase. Prior methods (Ren et al., 2024; Chen & Zhang, 2023; Lan et al., 2024; Chen et al., 2025) neglect spatial and temporal phase

coherence. Yet such coherence is essential for many RF sensing tasks including imaging and vital sign monitoring. In contrast, our simulator explicitly preserves phase information across space and time, supporting a wide range of phase-sensitive applications. Grounded in physical modeling, our approach generates high-fidelity signals directly, without requiring post-hoc learning-based signal refinement. We compare our phase-coherent ray tracing with existing ray tracing methods and observe significant improvements on phase-sensitive tasks, including circular beamforming imaging, respiration monitoring, and Doppler estimation, producing more faithful simulated signals and explicitly validating the accuracy of our phase modeling (§ 4.2). We further compare our simulated signals with real-world collected measurements for indoor scenes with moving persons, and simulations from a proprietary electromagnetic solver (Stolarski et al., 2018) for static environments. We find close agreement with 93.65% similarity on range-time spectrograms and 33.57dB PSNR on range-angle heatmaps, respectively, jointly validating the high fidelity and robustness of our simulations.

To demonstrate the effectiveness of WAVEVERSE, we conduct end-to-end evaluations on two case studies in RF imaging and human activity recognition, as detailed in § 4.3. When used as data augmentation by adding simulated samples to the training set, experiments show that WAVEVERSE consistently reduces depth prediction error and improves classification accuracy. Unlike prior methods, which often degrade or plateau as more simulated data is added, WAVEVERSE scales effectively, achieving up to an additional 18.12% MAE reduction in depth prediction and a 17.0 percentage improvement in classification accuracy.

## 2. Related Work

**RF Simulation.** Ray tracing has been widely used for radio propagation modeling, with early efforts addressing communication-centric applications such as signal coverage in static scenes (Yun & Iskander, 2015; Hoydis et al., 2023;

Yun & Iskander, 2024). For applications in RF sensing, prior work (Erol et al., 2020; Ahuja et al., 2021; Zhang et al., 2022; Xue et al., 2023) focuses on the signal interaction with human bodies neglecting environments and requires learning-based signal refinement. Some methods (Ren et al., 2024; Chen et al., 2025) use ray tracing for signal simulation but fall short of modeling spatial and temporal phase coherence. Inspired by the progress in image generation (Kingma et al., 2013; Goodfellow et al., 2020; Ho et al., 2020), data-driven methods (Chen & Zhang, 2023; Chi et al., 2024) combine ray tracing with neural networks for signal synthesis. However, they typically rely on large annotated datasets, offer limited controllability, lack physical interpretability, and provide no explicit modeling of multipath effects or signal phase coherence. Full-wave solvers like HFSS (Stolarski et al., 2018) provide accurate simulations but are computationally prohibitive for large-scale, dynamic indoor scenes. In contrast, our work develops a ray tracing framework with explicit spatial and temporal phase coherence, enabling high-fidelity RF simulation without any additional learning-based refinement.

**Human Motion Generation.** The generation of human motion has long been studied, with recent efforts focusing on how to enhance controllability. Text-based conditioning (Guo et al., 2022a;b; Zhang et al., 2023b; Tevet et al., 2023; Shafir et al., 2023; Jiang et al., 2023; Guo et al., 2024; 2025; Zheng et al., 2026) offers an intuitive interface but ignores environment context, yielding unrealistic movements under spatial constraints. To solve this issue, several methods (Tevet et al., 2023; Shafir et al., 2023; Wan et al., 2024; Xie et al., 2024; Dai et al., 2024) additionally introduce trajectories of explicit joint positions at designated frames. While effective, such time-indexed trajectories are over-constrained, as they require predefined durations, velocities, and careful alignment with text conditions, making the process labor-intensive, difficult to scale, and limited in generalization. Alternative approaches (Yi et al., 2024; Liu et al., 2024; Hwang et al., 2025) have explored motion generation directly within 3D scenes, but they either lack text conditioning capabilities or still require time-indexed inputs such as joint poses at specific frames and motion durations. Similar to trajectory-based methods, they impose substantial preparation overhead and ultimately limit the scalability of practical generation. Conversely, our path-based conditioning inherently addresses these issues and enables practical and scalable motion generation while providing high diversity and generalization.

## 3. Method

WAVEVERSE is an automated LLM-powered framework for simulating realistic RF signals in 3D indoor environments with human motions. As a prompt-driven framework,

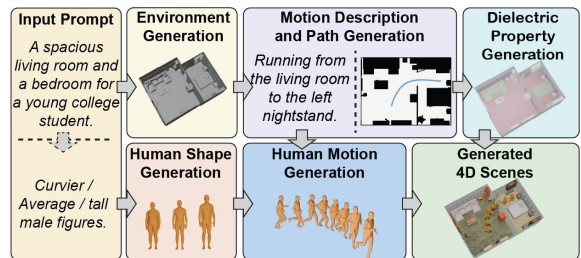

*Figure 2.* Overview of 4D World Generation.

WAVEVERSE can either be used interactively by a human user or fully automated by an LLM agent. Given a text description of an indoor environment, WAVEVERSE generates a text-aligned indoor environment with dynamic human activities, and finally simulates RF signals of the scene. This section describes the two core components of WAVEVERSE: (1) a 4D scene generator that synthesizes diverse indoor 4D scenes (§ 3.1), and (2) a phase-coherent ray tracing engine for signal simulation (§ 3.2).

### 3.1. 4D World Generation

WAVEVERSE utilizes a prompt-based pipeline to enable fully automated 4D world generation. Given a text description of the desired environment, whether provided by a user or an LLM, WAVEVERSE first constructs a semantically aligned 3D environment along with corresponding human body shapes. To generate realistic motion within a scene, WAVEVERSE first generates text descriptions and paths automatically, which are then used as conditions for our state-aware causal transformer for motion generation. In addition, to support realistic and physics-based RF simulation (§ 3.2), WAVEVERSE also assigns dielectric properties to scene objects with an LLM.

**3D Environment and Human Shape Generation.** WAVE-VERSE begins with a text description of the environment (Fig. 2). We build on an existing generation pipeline (Yang et al., 2024) to produce a structured layout, including floor plans, object categories, and placements, ultimately yielding a mesh representation of the indoor environment. This explicit 3D representation serves as a foundation for simulating RF signals as well as other modalities such as RGB images and depth maps. For human modeling, WAVEVERSE uses the SMPL model (Loper et al., 2023), a parametric human mesh that can be animated by adjusting pose and shape parameters. The shape parameters can be manually specified or automatically generated using a finetuned LLM (Árbol & Casas, 2024), which is conditioned on plausible body descriptions inferred from the input environment text.

**Motion Description and Path Generation.** To enable scalable motion generation for RF-based applications, WAVEVERSE animates SMPL with sequences of generated joint positions, referred to as motion in this paper. Our focus aligns

with RF tasks (Singh et al., 2019; Pan et al., 2024), requiring diverse whole-body dynamics rather than object-centric interactions or simple locomotion. The key challenge is to generate human motion that matches the semantic context of the environment while ensuring diversity and spatial realism, such as avoiding wall penetrations. One approach to achieve such control is to pair text descriptions with time-indexed *trajectory* inputs, that is, precise 3D joint positions specified at key frames, consistent with the text and environment. However, it requires careful alignment with text and extensive manual specification, making it labor-intensive and difficult to scale. Moreover, by fixing joint positions, durations, and velocities in advance, it effectively predetermines the motion and reduces flexibility, limiting generalization.

To address this challenge, we decompose motion generation into two stages. Given a text prompt of the environment, an LLM first produces a motion description, like "wave the arm", and specifies the start and end 2D positions on the floor, which can also be provided by users. We replace trajectory constraints with *paths*, a set of $L$ spatial waypoints that guide where the person should move without prescribing velocity or duration. Such paths can be readily generated with path-finding algorithms given start and end points. For model training, we derive paths by downsampling and projecting the pelvis trajectory from motion sequences into $L=64$ spatially evenly spaced 2D waypoints. We delegate the motion generation task to later models, while LLMs focus on high-level reasoning.

**Conditional Human Motion Generation.** The second step is to generate motion sequences conditioned on the input texts and paths. Since the path does not specify motion duration, we adopt an autoregressive model that dynamically determines when to terminate the sequence, unlike existing methods that generate human motion with a predefined duration (Tevet et al., 2023; Xie et al., 2024; Dai et al., 2024). Specifically, motion sequences are first tokenized using VQ-VAE (Van Den Oord et al., 2017), achieving motion tokens $X = [m_1, m_2, \ldots, m_n, m_{\text{end}}]$, where $m_i \in \{1, \ldots, M\}$ indexes a learned codebook, and $m_{\text{end}}$ denotes the end of the sequence. The motion description is encoded using CLIP (Radford et al., 2021), while the 2D waypoint sequence is processed through an MLP-based position encoder, producing condition embeddings $c = (c_{\text{text}}, c_{\text{path}_0}, \ldots, c_{\text{path}_L})$.

While existing autoregressive models (Zhang et al., 2023b) generate motion via next-token prediction, learning the distribution $P(m_n \mid c, m_0, \ldots, m_{n-1})$, we find this formulation struggles to align motion with the input path. Inspired by reinforcement learning (Kaelbling et al., 1996), we view next-token prediction as a sequential decision-making process, where each token is an action. We argue that the absence of explicit spatial context at each decision step limits

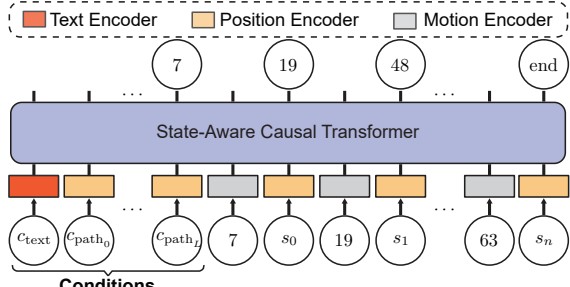

*Figure 3.* State-Aware Causal Transformers.

path adherence. To address this, we introduce a *state-aware causal transformer* (Fig. 3), conditioning each prediction on the current spatial state. Formally, the next-token distribution is modeled as $P(m_n \mid c, m_0, s_0, \ldots, m_{n-1}, s_{n-1})$, where $s_i$ encodes the 2D position at the last frame up to token $m_i$, with the same position encoder.

Despite the benefits of spatial state conditioning, we observe the model overfits by relying heavily on path information, resulting in poor text alignment. To mitigate this and promote balanced conditioning, we introduce a path-masking strategy during training. We first uniformly sample a masking ratio $r \in [r_{\min}, r_{\max}]$ to set the target number of waypoints to mask. We then iteratively select and mask random contiguous segments of length up to $\ell$, until the target ratio is reached. We find that this sequential masking strategy improves generalization and enhances text-motion alignment (§ 4.1).

**Dielectric Property Generation.** To enhance physical realism, WAVEVERSE adopts the frequency-dependent parametric model (Series, 2015) for dielectric properties, using validated permittivity and conductivity parameters for 14 common materials. To expand this set, we prompt an LLM to propose additional material categories and corresponding model parameters, retaining only those with dielectric values within documented physical ranges, yielding a curated library of 24 materials. Material categories are preassigned to objects by the LLM and then retrieved in scene generation, ensuring physically grounded dielectric values while allowing semantic descriptions to guide material selection.

### 3.2. RF Signal Simulation

Given the generated 4D scenes, WAVEVERSE employs ray tracing to simulate RF signals. Existing RF ray tracing engines, however, inherit practices from computer graphics, where the focus is on modeling signal amplitude and rays are cast stochastically (Cook, 1986; Nimier-David et al., 2019). As a result, RF simulation similarly casts rays randomly over a spherical or conical distribution, resulting in inconsistent ray-surface interactions across frames and radar positions (Chen & Zhang, 2023; Ren et al., 2024). This in-

consistency poses significant challenges for RF applications, where signal phase plays a critical role. For example, RF imaging distinguishes objects at the same range but different angles by leveraging phase differences through beamforming. Similarly, Doppler-based velocity estimation relies on phase shifts across chirps caused by object motion. To address this, we introduce *phase-coherent ray tracing* that operates on the scene mesh and ensures consistent ray-surface interactions across different radar positions and over time as objects move. It preserves signal phase coherence, enabling accurate simulation of phase-dependent RF phenomena.

**RF Simulation with Ray Models.** Ray tracing models wave propagation as a collection of discrete paths connecting the transmitter (Tx) and receiver (Rx) in the scene. Let $\{\mathcal{P}_k\}_{k=1}^K$ denote the set of valid propagation paths identified by ray tracing. Each path $\mathcal{P}_k$ is characterized by four parameters: a propagation delay $\tau_k$; a complex coefficient $a_k$, whose magnitude encodes attenuation due to path loss and interactions with scene surfaces, and whose angle represents accumulated phase shifts; an angle of departure (AoD) $\theta_k$ at Tx; and an angle of arrival (AoA) $\varphi_k$ at Rx. The channel impulse response (CIR) $h(t)$, which describes how an impulse propagates from Tx to Rx, is modeled as the superposition of all paths: $h(t) = \sum_k a_k \cdot G_{\text{Tx}}(\theta_k) \cdot G_{\text{Rx}}(\varphi_k) \cdot \delta(t - \tau_k)$, where $G_{\text{Tx}}$ and $G_{\text{Rx}}$ denote the antenna gain patterns of the transmitter and receiver, capturing their directionality, and $\delta(t)$ is the Dirac delta function. Any signal received by Rx can then be computed as the convolution between the transmitted signal and the CIR.

**Phase-Coherent Ray Tracing.** As discussed, conventional ray tracing methods fall short in preserving phase coherence, as they cast rays stochastically, resulting in different ray-surface interactions even for nearby radar positions (Fig. 4(a), left). This issue becomes more severe in dynamic scenes with moving humans, where changes in geometry cause rays to strike entirely different surface points across frames, breaking temporal phase coherence (Fig. 4(b)).

To overcome these challenges, we propose phase-coherent ray tracing that ensures consistent ray-surface interactions across space and time. To achieve *spatially-coherent ray tracing*, i.e., ensuring coherent phase variation across different radar locations, we generate paths for each radar from a fixed set traced from a representative reference radar. Specifically, assume we are synthesizing signals for $N$ radars with poses $\{(\mathbf{t}_n, \mathbf{r}_n)\}$ for $n = 1, \ldots, N$, where $\mathbf{t}_n$ and $\mathbf{r}_n$ denote the position and rotation of the transmitter and receiver. We define a reference $(\mathbf{t}_0, \mathbf{r}_0)$ as the geometric center of all radar positions, and trace rays uniformly over a sphere, to obtain paths $\{\mathcal{P}_k\}$ between $\mathbf{t}_0$ and $\mathbf{r}_0$. For each path $\mathcal{P}_k$, we represent it with a sequence of 3D points $\mathcal{P}_k = [\mathbf{t}_0, \mathbf{p}_1, \cdots, \mathbf{p}_{D_k}, \mathbf{r}_0]$ where $\mathbf{p}_d$ denotes the $d$-th surface interaction point along the path and $D_k$ denotes the

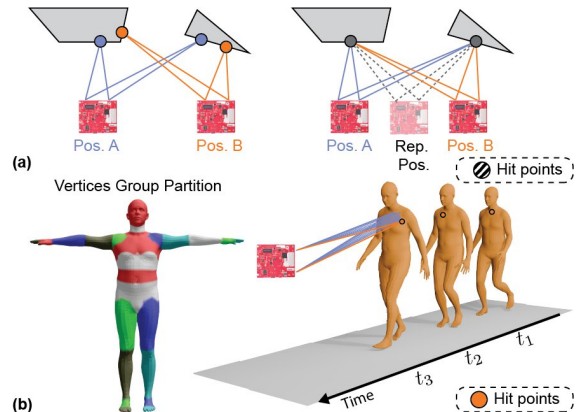

*Figure 4.* Illustration of phase-coherent ray tracing. (a) shows spatial coherence by tracing consistent paths across two radar locations. (b) depicts temporal coherence for a moving person at timestamps $t_1$, $t_2$, and $t_3$. For clarity, rays at $t_2$ and $t_3$ are omitted.

number of encountered surfaces for this path.

To generate paths for a radar with poses $(\mathbf{t}_n, \mathbf{r}_n)$, we modify each reference path $\mathcal{P}_k$ by replacing the original transmitter and receiver $(\mathbf{t}_0, \mathbf{r}_0)$ with the current ones, as shown in the right side of Fig. 4(a). We then compute the CIR for each modified path using updated propagation delay, attenuation, phase, AoD, and AoA. Occlusion checks are further performed on the resulting paths, and blocked paths are discarded. By preserving consistent surface interaction points, our approach ensures spatial phase coherence across radars with various poses while avoiding redundant ray tracing.

To obtain *temporally-coherent ray tracing*, i.e., coherent phase changes as humans move within the scene, we remap ray-surface interactions from individual vertices to semantically or spatially coherent groups. While ray tracing is performed independently over time, we enable temporal coherence by expanding ray hits over stable vertex groups that persist across frames. Specifically, we partition all vertices $\mathcal{V} = \{\mathbf{v}_m\}_{m=1}^M$ of a human mesh into $G$ disjoint, semantically coherent groups $\{\mathcal{V}_g\}_{g=1}^G$, where $\cup_g \mathcal{V}_g = \mathcal{V}$, with a grouping function $\mathcal{G} : \mathcal{V} \to \{1, \ldots, G\}$. At each timestamp $t$, ray tracing yields a set of paths $\{\mathcal{P}_k^{(t)}\}$. When a path $\mathcal{P}_k^{(t)}$ intersects the human mesh at point $\mathbf{p}_d^{(t)}$, we associate it with a representative vertex $\hat{\mathbf{p}}_d^{(t)}$, a fixed vertex of the intersected face, to enable consistent grouping. We then expand the path by replacing the hit point with all vertices within the same group, i.e., those satisfying $\mathcal{G}(\mathbf{v}_m) = \mathcal{G}(\hat{\mathbf{p}}_d^{(t)})$. Notably, we first replace $\mathcal{P}_k^{(t)}$ with the following set of paths: $[\mathbf{t}_0, \ldots, \mathbf{p}_{d-1}^{(t)}, \mathbf{v}_m, \mathbf{p}_{d+1}^{(t)}, \ldots, \mathbf{r}_0]$, where $\mathcal{G}(\mathbf{v}_m) = \mathcal{G}(\hat{\mathbf{p}}_d^{(t)})$ for $m \in \{1, \ldots, M\}$. We then perform occlusion checks on the expanded paths and denote the number of valid paths as $N_{\text{valid}}$. For each valid path, we compute propagation delay, attenuation, phase, AoD and AoA. To conserve overall

*Table 1.* Text and path conditioned motion generation performance. **Bold** for the best and underline for the second best.

| Method | Architecture | R-Prec. ↑ | FID ↓ | Div. → | Path Error ↓ | | Ending Error ↓ | |
|---|---|---|---|---|---|---|---|---|
| | | | | | > 20 cm | > 60 cm | > 20 cm | > 60 cm |
| Ground Truth | | 0.797 | 0.002 | 9.503 | 0. | 0. | 0. | 0. |
| MDM | Diffusion | 0.719 | 0.295 | **9.462** | 0.547 | 0.207 | 0.666 | 0.367 |
| OmniControl | Diffusion | 0.751 | 0.319 | 9.279 | 0.239 | 0.083 | 0.330 | 0.152 |
| MotionLCM | Diffusion | 0.739 | 0.754 | 9.588 | 0.315 | 0.055 | 0.468 | 0.177 |
| T2M-GPT | Autoregressive | 0.691 | 0.377 | 9.736 | 0.406 | 0.127 | 0.545 | 0.255 |
| Ours | Autoregressive | **0.755** | **0.238** | 9.445 | **0.208** | **0.045** | **0.325** | **0.111** |

signal energy, attenuation is further divided by $N_{\text{valid}}$. In practice, we expand only the first hit point from the transmitter, both to avoid exponential growth from higher-order reflections and because single-bounce paths typically dominate received energy due to lower propagation loss.

**Flexible Configuration.** WAVEVERSE generalizes to a wide range of radar configurations, including arbitrary antenna positions and orientations, gain patterns, frequency bands, and sampling rates, making it adaptable to diverse hardware setups. This flexibility stems from our CIR modeling, which naturally accounts for these factors and applies consistently across configurations. By simulating received signals through convolution with the transmitted waveform, WAVEVERSE supports diverse RF protocols. Additionally, relying on explicit physical modeling, WAVEVERSE scales to unseen conditions while achieving accurate and reliable signal behavior, offering robustness and scalability that are difficult to achieve with data-driven methods.

## 4. Experiments

In this section, we evaluate the 4D world generation and the signal simulation in WAVEVERSE. We begin with benchmarks and ablation studies of the proposed state-aware causal transformer for text and path-conditioned motion generation, and analyze the generated 4D world. We then evaluate the proposed phase-coherent ray tracing. Finally, we assess WAVEVERSE in two real-world case studies.

### 4.1. Performance of Human Motion Generation

**Dataset and Evaluation Metrics.** We evaluate models over the HumanML3D (Guo et al., 2022a) dataset for benchmarks. It contains 14,616 captioned human motion sequences. In all experiments, we fix $L = 64$. More details can be found in Appendix A.1.1. Following the evaluation protocol from OmniControl (Xie et al., 2024), we report *R-Precision* to quantify the text-motion alignment, the *Frechet Inception Distance (FID)* to assess motion quality, and the *Diversity* score to measure the variability of the generated motion. To assess spatial alignment with the path condition, we define the *Path Error* as the average per-point $L_2$-

distance between the generated path and the ground-truth path, and *Ending Error* as the deviation at the final timestamp, capturing how closely the generated motion reaches the target endpoint. For both metrics, we report three statistics, including the mean error and the fractions of samples exceeding 20 cm and 60 cm.

**Comparison to Baselines.** We adopt four open-source, state-of-the-art motion generation methods as baselines, selected as the closest in design to ours: the diffusion-based MDM (Tevet et al., 2023), OmniControl (Xie et al., 2024) and MotionLCM (Dai et al., 2024), and the autoregressive T2M-GPT (Zhang et al., 2023b). Details of the model adaptations can also be found in Appendix A.1.1.

As shown in Tab. 1, our method consistently outperforms all baselines across R-Precision, FID, and path-following metrics, demonstrating better motion quality and alignment with input conditions. MDM relies solely on global path features as conditioning, which provide limited guidance for path adherence. OmniControl loses localized per-frame control when adapted for paths and becomes unstable with its analytic optimization, achieving a high fraction of large-error samples. While explicit path supervision in MotionLCM reduces large path deviations, the overall performance remains moderate across metrics. We provide a detailed analysis in Appendix A.1.2, and emphasize that our improvements stem from the proposed modules rather than the autoregressive formulation itself. Importantly, T2M-GPT, on which our method is built, underperforms diffusion-based baselines, whereas our approach achieves better performance. We also present qualitative comparisons in Appendix A.1.3.

**Ablation Studies.** We validate the key components of the state-aware causal transformer introduced in § 3.1 through three ablation studies with mean path/ending errors for path adherence evaluation, summarized in Tab. 2. First, without path masking, the model overfits to path conditions, resulting in degraded motion quality and worse text alignment, evidenced by higher FID and lower R-Precision. Path masking alleviates this issue and also improves generalization in path following. We explore alternative approaches in Appendix A.1.4, but they only achieve suboptimal performance.

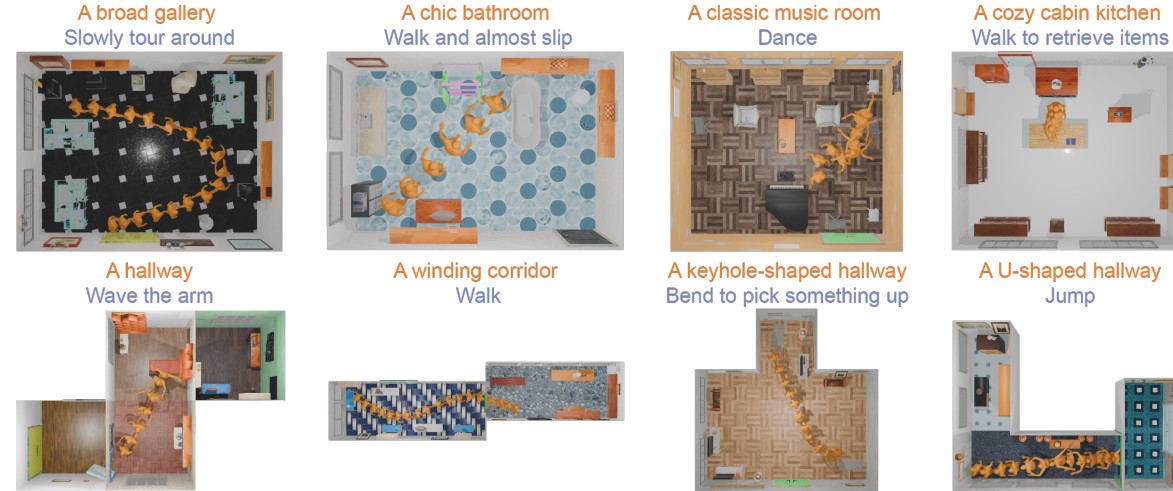

*Figure 5.* Visualization of generated 4D scenes, together with the environment prompts and motion descriptions used for generation.

*Table 2.* Ablation study for components and hyperparameters.

| Setting | Model | R-Prec. ↑ | FID ↓ | Path Err. ↓ | Ending Err. ↓ |
|---|---|---|---|---|---|
| **Components** | Ours | 0.755 | 0.238 | 0.151 | 0.287 |
| | w/o Mask | 0.643 | 0.747 | 0.192 | 0.325 |
| | w/o State | 0.757 | 0.422 | 0.250 | 0.460 |
| | w/o Both | 0.691 | 0.377 | 0.274 | 0.528 |
| **Masking Rate** | [0.5, 0.9] | 0.755 | 0.238 | 0.151 | 0.287 |
| | [0.1, 0.5] | 0.691 | 0.396 | 0.171 | 0.312 |
| | [0.1, 0.9] | 0.713 | 0.298 | 0.160 | 0.303 |
| **Segment Length** | 5 Points | 0.755 | 0.238 | 0.151 | 0.287 |
| | 10 Points | 0.763 | 0.342 | 0.207 | 0.393 |
| | 15 Points | 0.776 | 0.339 | 0.228 | 0.403 |

We also show that removing state information markedly reduces path-following capability, underscoring its importance. Second, for the masking rate range $[r_{\min}, r_{\max}]$, our choice $[0.5, 0.9]$ outperforms both $[0.1, 0.5]$ and $[0.1, 0.9]$, showing that higher rates better balance reliance on path and text. Finally, varying contiguous masking length $\ell$ reveals a trade-off. Shorter segments (5 points) enhance path alignment and lower FID, whereas longer ones (10 or 15 points) improve text alignment but substantially degrade path following. We therefore adopt 5 points in our model.

**Scene Evaluation.** To evaluate the generated scenes, we analyze both the quality and diversity of the generated results, which couple indoor environments with human motion. Across 120 trials, our pipeline achieves a 95.83% success rate, yielding 115 unique environments that cover a wide range of room types, object arrangements, and material properties, with two motion sequences synthesized for each scene. The remaining failures mainly arise from floor-plan errors during environment generation or overly constrained layouts that prevent feasible path construction. In total, the dataset contains 920 unique object assets, 47 room categories, and 24 dielectric materials, with an aver-

age of 25 object instances per scene and an average motion duration of 4.5 seconds. To assess spatial compatibility between motion and scene geometry, we report a collision ratio of 2.35%, defined as the fraction of motion frames that collide with the environment. We also measure cumulative collision depth, which averages 12.23 cm and reflects the accumulated interpenetration over each motion sequence. These results indicate that the generated motions conform well to their surrounding environments. Qualitative results in Fig. 5 show environment-motion pairs with varied layouts and objects. Additional visualizations are provided in Fig. 15 in Appendix A.2.1, with execution-time details reported in Appendix A.2.2.

### 4.2. Performance of Phase-Coherent Ray Tracing

To evaluate the effectiveness of our phase-coherent ray tracing, we conduct three benchmarks that rely on accurate phase modeling. The baseline follows standard ray tracing methods (Ren et al., 2024; Chen et al., 2025). Our method differs only by incorporating spatial and temporal phase coherence, ensuring fair comparison. For fidelity evaluation, we also compare simulated signals against real measurements and Ansys HFSS (Stolarski et al., 2018) simulations.

**Spatial Phase Coherence.** To evaluate spatial phase coherence, we adopt the same panoramic imaging setup as in Lai et al. (2024), combining signals from 1,200 radar positions and orientations arranged along a circular path. Fig. 6 shows the imaging results from two random environments generated with beamforming. The improved image clarity highlights the role of spatial phase coherence, ensuring that wavefronts remain coherently aligned across all radar poses. Notably, the ghost reflections in the imaging result also highlight that our simulation captures multipath effects that are difficult to guarantee in purely learning-based meth-

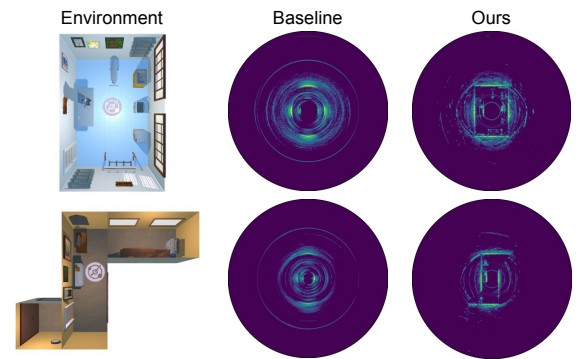

Figure 6. Panoramic imaging w/ and w/o spatial phase coherence.

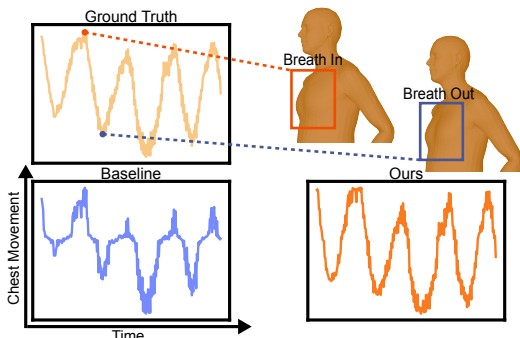

Figure 7. Recovered motion w/ and w/o temporal phase coherence.

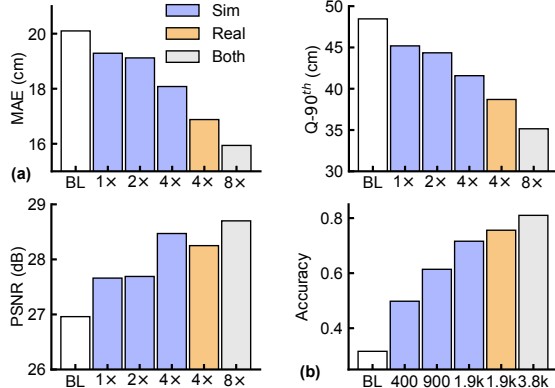

Figure 8. Performance comparison over the baseline with varying amounts of added real and simulated data on: (a) RF imaging and (b) human activity recognition. BL: Baseline.

ods. These results jointly demonstrate the importance of spatial phase coherence for downstream RF sensing applications. More visualizations and comparisons are provided in Appendix A.3.1.

**Temporal Phase Coherence.** As discussed in § 3.2, stochastic ray casting fails to preserve phase in dynamic settings, making it fundamentally unsuitable. For the baseline, we therefore adopt a minimal modification by fixing the rays cast across time, though it deviates from standard practice. We validate temporal coherence on a respiration tracking task by animating the SMPL with real breathing signals from Li et al. (2024), generating 500 seconds of data across 40 sequences. As the chest moves, minute chest-to-radar distance changes are captured in the phase (Zhao et al., 2016). We extract this phase from the simulated signals and convert it into distance change. With temporal coherence, the reconstructed curves achieve 0.08 RMSE and 8.89 DTW against ground truth, significantly outperforming the baseline at 0.14 and 12.68. We further simulate a sinusoidally moving sphere and generate range–Doppler heatmaps via range and Doppler FFT, where our approach again outperforms the baseline. We provide qualitative results in Appendix A.3.2.

**Comparison with Real Measurements and HFSS Simulation**. To explicitly evaluate the fidelity of the simulated signals, we compare WAVEVERSE against both real-world measurements and high-fidelity electromagnetic simulations.

In the real-world setup, a subject walks in front of a wall, and motion is reconstructed from cameras using WHAM (Shin et al., 2024). The surrounding scene geometry and material properties are also recovered. WAVEVERSE achieves 28.63dB PSNR and 93.65% energy similarity when compared to the measured range–time spectrograms. To provide a broader validation, we additionally assess signal fidelity across diverse environmental setups and signal configurations, and analyze the impact of material modeling. These studies provide complementary evidence for the fidelity and robustness of WAVEVERSE under varied scene and sensing conditions, with detailed results provided in the Appendix. We further compare against Ansys HFSS (Stolarski et al., 2018), a proprietary, industry-grade EM solver that is commercially licensed. Across 16 indoor setups, WAVEVERSE closely matches HFSS outputs, achieving 33.57dB PSNR with only a 2.12% normalized RMSE on average. While HFSS typically requires over an hour for a single simulation, WAVEVERSE achieves comparable results in well under one second. Further details can be found in Appendix A.3.3.

### 4.3. Case Studies

Having established the effectiveness of individual components, we now evaluate the full pipeline of WAVEVERSE in real-world scenarios. To this end, we conduct two case studies on RF-based applications with publicly available data: RF imaging (Lai et al., 2024) and human activity recognition (Singh et al., 2019). We evaluate each task under two conditions: a limited-data setting that reflects practical constraints, and a data-rich setting where more real-world samples are available, and compare with prior work to highlight the advantages of our approach.

**RF Imaging.** We evaluate depth prediction from RF signals with the ML model of Lai et al. (2024), which employs a rotating radar setup adopted in § 4.2 and remains robust under low visibility and harsh weather. We apply the same

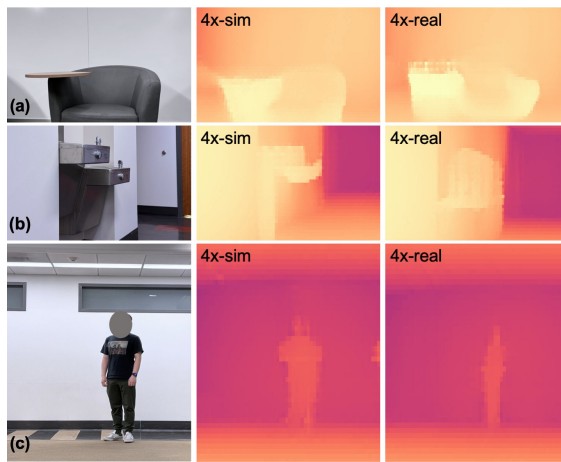

*Figure 9.* Improved imaging quality over objects. (a) Armchair. (b) Water fountain. (c) Human.

*Table 3.* Comparison with Standard RT on RF Imaging.

| Metric | Method | Real only | +1× sim | +2× sim | +4× sim |
|---|---|---|---|---|---|
| MAE (↓) | WAVEVERSE | 20.10 | 19.29 | 19.12 | 18.08 |
| | Standard RT | 20.10 | 21.45 | 21.89 | 22.28 |
| Q-90th (↓) | WAVEVERSE | 48.46 | 45.19 | 44.35 | 41.58 |
| | Standard RT | 48.46 | 49.98 | 50.24 | 53.29 |
| PSNR (↑) | WAVEVERSE | 26.96 | 27.66 | 27.69 | 28.47 |
| | Standard RT | 26.96 | 27.01 | 26.85 | 26.89 |

*Table 4.* Comparison with RF Genesis on Activity Recognition.

| Method | Real only | + 4× sim | + 9× sim | + 19× sim |
|---|---|---|---|---|
| WAVEVERSE | 31.6% | 49.8% | 61.4% | 71.6% |
| RF Genesis | 31.6% | 46.6% | 55.8% | 54.6% |

cross-building protocol, where the model is trained on RF data from 11 buildings and evaluated on 1,000 frames from a held-out building. To improve prediction under limited data, we augment with simulated RF signals and depth supervision generated by WAVEVERSE. We sample 1,000 real frames as a baseline dataset and progressively add 1×, 2×, and 4× simulated data from 115 diverse scenes in § 4.1.

Fig. 8(a) shows consistent improvements in MAE, 90th percentile error, and PSNR as simulated training data increases, outperforming the baseline trained on limited real data. With 4× simulated data, MAE and 90th percentile error drop by 2.02 cm and 6.88 cm, while PSNR improves by 1.51 dB. The gains show that simulated data alone can enhance performance in data-limited settings. For comparison, we include a 4×-real setting trained with 4,000 additional real samples. Notably, the simulated data captures 73.33% of the improvement in 90th percentile error, and surpasses it in PSNR. Our analysis shows the model excels in high-quality ranges. 12.1% of predictions exceed 35 dB, nearly double the 6.6% baseline. For the broader 30 dB threshold, the proportion rises from 41.9% to 45.4%. We visualize improvements in imaging over objects in Fig. 9 and attribute the gains to the rich object diversity in our scenes. Finally, combining simulated and real data yields the best performance, with an additional gain of 3.55 cm in 90th percentile error and 0.45 dB in PSNR, highlighting the value of WAVEVERSE-generated signals in both limited and rich data scenarios.

To further demonstrate the benefits of WAVEVERSE-generated signals, we compare them to a Standard Ray Tracing (RT) baseline (Ren et al., 2024; Chen et al., 2025) using the same augmentation setup. Learning-based methods like RF Genesis (Chen & Zhang, 2023) are not applicable due to fixed radar assumptions and a lack of support for continuous rotational motion. As shown in Tab. 3, WAVEVERSE consistently outperforms Standard RT across all metrics

and augmentation levels. Its performance improves steadily with more synthetic data, while adding Standard RT data yields no gain, suggesting that traditional simulations lack the realism needed to facilitate learning and generalization.

**Human Activity Recognition.** We further evaluate WAVEVERSE on an open-source, privacy-preserving human activity classification task (Singh et al., 2019), which maps RF signal sequences to activities. To synthesize motions, we use an LLM to generate diverse descriptions for the five activities in the dataset: walking, standing, squatting, jumping, and jumping jacks. A classifier trained on 100 real samples and tested on 500 held-out samples achieves a baseline accuracy of 31.6% (Fig. 8(b)). Augmenting with 400, 900, and 1900 simulated samples progressively improves accuracy to 49.8%, 61.4%, and 71.6%, approaching the 75.6% from training on all 2,000 real samples. Combining all simulated and real data yields the best performance of 81.0%.

We also compare WAVEVERSE with RF Genesis (Chen & Zhang, 2023). Tab. 4 shows results when augmenting real samples with 4×, 9×, and 19× simulated data. WAVEVERSE consistently delivers larger accuracy gains across all levels. While RF Genesis shows some improvement at low augmentation ratios, its performance plateaus as more data is added. In contrast, WAVEVERSE continues to scale, highlighting the benefits of its physically grounded simulation for generating high-fidelity signals.

## 5. Conclusion

We present WAVEVERSE, a prompt-based, scalable framework that generates dynamic 4D environments with human motion and simulates realistic RF signals via phase-coherent ray tracing. Comprehensive evaluations and case studies demonstrate the practical utility of WAVEVERSE in enabling high-fidelity RF data generation and enhancing performance in both data-limited and data-rich scenarios. We release our code and simulator to support future research.

## Impact Statement

In this paper, we introduce WAVEVERSE, a scalable and physically grounded framework for simulating RF signals in dynamic environments. The ability to generate realistic RF data with diverse human motion and scene layouts has several potential positive societal impacts. It can facilitate progress in privacy-preserving sensing, indoor navigation, and health monitoring by reducing reliance on vision-based sensors. By supporting high-fidelity simulation under varied conditions, WAVEVERSE may also help broaden access to RF research, lowering the barrier to entry for institutions without expensive hardware or large-scale data collection pipelines. However, WAVEVERSE may also entail potential negative societal impacts. Since WAVEVERSE relies on LLMs to generate human motions and semantic scene layouts, it inherits the risks associated with LLMs, such as biases in generated content and unintended reinforcement of stereotypes, which users should pay attention to.

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

# A. Appendix

In this appendix, we first present additional details and results of WAVEVERSE from three perspectives, conditional human motion generation, 4D world generation, and RF simulation, organized consistently with the structure of the main paper, and then discuss the limitations and future work. We also include videos of qualitative results on our webpage and recommend viewing them for a clearer demonstration.

## A.1. Conditional Human Motion Generation

### A.1.1. DETAILS

**Model Details.** Our model comprises two main components: a VQ-VAE (Van Den Oord et al., 2017) tokenizer and the proposed state-aware causal transformer. The VQ-VAE is built with 1D convolution layers, residual blocks, and ReLU activations in both the encoder and decoder along the temporal dimension. It applies a temporal downsampling rate of 4 and uses a codebook of size $512 \times 512$. The tokenizer is trained for 300K iterations with a batch size of 256 using the AdamW optimizer (Loshchilov & Hutter, 2017) ($\beta_1 = 0.9$, $\beta_2 = 0.99$). Following Van Den Oord et al. (2017); Zhang et al. (2023b), the training objective combines reconstruction, embedding, and commitment losses. To enhance motion quality and training stability, we adopt velocity regularization, exponential moving average updates, and codebook resetting as in Zhang et al. (2023b). The learning rate is initialized at 2e-4 and decayed by a factor of 0.05 after 200K iterations with a MultiStepLR scheduler.

The state-aware causal transformer consists of 8 transformer layers (Vaswani et al., 2017), each with 8 attention heads and a hidden dimension of 512. Temporal causality is enforced by applying causal self-attention (Radford et al., 2018) across the network. Text conditions are encoded with CLIP (Radford et al., 2021), while path conditions and spatial states are encoded by a 3-layer MLP with a hidden dimension of 256. The model is trained to maximize the likelihood of token sequences using cross-entropy loss with a batch size of 128. Optimization is performed using the Adam optimizer ($\beta_1 = 0.5$, $\beta_2 = 0.9$) for 300K iterations. The learning rate is initialized at 1e-4 and decayed by a factor of 0.05 after 150K iterations using a MultiStepLR scheduler.

**Dataset and Baseline Details.** We adopt HumanML3D (Guo et al., 2022a) as our dataset, which contains 14,616 motion sequences annotated with 44,970 text descriptions. To extract path information, we downsample the pelvis trajectory into 64 spatially evenly spaced 2D waypoints on the floor, which serve as the path condition. Notably, the path encodes only directional guidance and excludes duration or velocity information. The dataset is split following the standard protocol as that in Tevet et al. (2023); Xie et al. (2024); Zhang et al. (2023b); Dai et al. (2024).

We adopt four open-source, state-of-the-art motion generation methods as baselines, selected as the closest in design to ours: the diffusion-based MDM (Tevet et al., 2023), OmniControl (Xie et al., 2024) and MotionLCM (Dai et al., 2024), and the autoregressive T2M-GPT (Zhang et al., 2023b). MDM, OmniControl, and MotionLCM support trajectory-conditioned motion generation, which is close to our path-conditioned framework, whereas T2M-GPT serves as the base model for our approach. For MDM, OmniControl, and MotionLCM, we follow their original setup, providing target motion length during both training and inference. T2M-GPT dynamically determines when to terminate the sequence by outputting an [end] token. To incorporate path conditioning, we apply only the necessary modifications while keeping all other components unchanged, as described below.

For MDM, we incorporate path conditions by adding the encoded path features to its original conditioning inputs. For OmniControl, we make a minimal change by replacing the per-frame joint encodings with a shared global path feature that is applied uniformly to all joints. We adopt the MLP design as our method does for a fair comparison. We experimented with both max pooling and mean pooling for aggregating path features, and found that max pooling consistently yields better performance. Thus, we apply max pooling when encoding paths for MDM and OmniControl. In addition, we retain the spatial guidance of OmniControl by similarly applying an analytic function that evaluates how closely the generated motion path aligns with the desired path. The gradient of this function is then used to explicitly perturb the predicted mean at each denoising step, guiding the generated motions to follow the specified path. For MotionLCM, we preserve its stacked transformer layers to encode path signals, as originally designed for trajectory encoding, and leverage the extracted features in the same way as in the original implementation. We also retain its original trajectory-alignment loss but apply it with paths, explicitly penalizing deviations between the generated and desired paths during training. For T2M-GPT, we extend the input sequence by appending path tokens after the text tokens, mirroring our own path condition encoding to ensure fairness in comparison. All other settings, including hyperparameters, follow the original configurations reported in the

*Table 5.* Text and path conditioned motion generation performance. **Bold** for the best and underline for the second best. R-Prec.: R-Precision; Div.: Diversity.

| Method | Architecture | R-Prec. ↑ | FID ↓ | Div. → | Path Error ↓ | | Ending Error ↓ | |
| --- | --- | --- | --- | --- | --- | --- | --- | --- |
| | | | | | > 20 cm | > 60 cm | > 20 cm | > 60 cm |
| Ground Truth | | 0.797 | 0.002 | 9.503 | 0. | 0. | 0. | 0. |
| MDM | Diffusion | 0.719 | 0.295 | **9.462** | 0.547 | 0.207 | 0.666 | 0.367 |
| OmniControl | Diffusion | 0.751 | 0.319 | 9.279 | 0.239 | 0.083 | 0.330 | 0.152 |
| MotionLCM | Diffusion | 0.739 | 0.754 | 9.588 | 0.315 | 0.055 | 0.468 | 0.177 |
| T2M-GPT | Autoregressive | 0.691 | 0.377 | 9.736 | 0.406 | 0.127 | 0.545 | 0.255 |
| Ours | Autoregressive | **0.755** | **0.238** | 9.445 | **0.208** | **0.045** | **0.325** | **0.111** |

respective papers. All models are implemented in PyTorch and trained on an NVIDIA L40 GPU.

### A.1.2. COMPARISON TO BASELINES

As shown in Tab. 5, our method consistently outperforms all baselines across R-Precision, FID, and path-following metrics, demonstrating superior motion quality and alignment with input conditions. MDM supports trajectory-conditioned motion generation by formulating it as an inpainting task. However, it explicitly leverages known keyframes during denoising, which are unavailable in our path-conditioned framework, and it lacks explicit mechanisms to guide or evaluate motions against the desired path, ultimately limiting its ability to satisfy path conditions. For OmniControl, its original per-frame joint control signals are replaced with global path conditions, which removes localized frame-wise guidance. Furthermore, its analytic path function computes a weighted sum over joint positions, where the weights can vary across denoising steps, leading to instability and higher rates of large path-following errors, despite competitive accuracy at lower thresholds. MotionLCM leverages a path-supervision loss between the predicted and ground-truth paths, which reduces high-level path errors but still results in only moderate performance overall.

In contrast, our method generates motions in a stable, end-to-end autoregressive manner with spatial state feedback, enabling precise and controllable motion generation. Importantly, it does not require predefined duration, making it more scalable in practice. We also emphasize that the gains arise from our proposed modules rather than from the autoregressive structure itself. Crucially, T2M-GPT, on which our method is built, underperforms diffusion-based baselines, whereas our approach achieves better performance. This highlights the effectiveness of our proposed modules, validated by the ablation study in § 4.1.

To assess physical plausibility, we additionally report two metrics: *Skating Ratio* for foot sliding and *Bone-Length Variance* for skeletal stability. Our model achieves a skating ratio of 0.067, closely matching 0.057 for real data, and a bone-length variance of 1.78 cm$^2$, indicating stable limb lengths. These results show that our motions are not only text and path aligned, but also physically plausible and faithful to human dynamics.

### A.1.3. QUALITATIVE RESULTS

Additionally, we present qualitative results of our method for text and path conditioned human motion generation. We begin with customized conditions to highlight the capabilities of our model, followed by qualitative results from the test set of the HumanML3D dataset.

We begin by presenting qualitative results that demonstrate the model's ability to follow diverse path lengths. To this end, we use the same text condition, *walk*, while varying paths of lengths [1, 3, 5, 7] meters, all oriented in the same direction. As shown in Fig. 10, the input paths are visualized with colored points transitioning from **blue** to **red** to indicate temporal order. The generated motions closely follow the given paths while remaining consistent with the texts.

We then change the text prompt to *slowly walk* and fix both the text and path length while varying the path direction by angles of $\pm 90°$, $\pm 45°$, and $\pm 30°$. The corresponding visualizations are shown in Fig. 11. The generated motions exhibit slower velocities compared to those in previous examples, reflecting the semantics of the updated text conditions. We refer the reviewer to the video on our webpage for a clearer comparison. Additionally, the visualizations show that the generated motions accurately follow paths with varying directions, demonstrating strong path adherence.

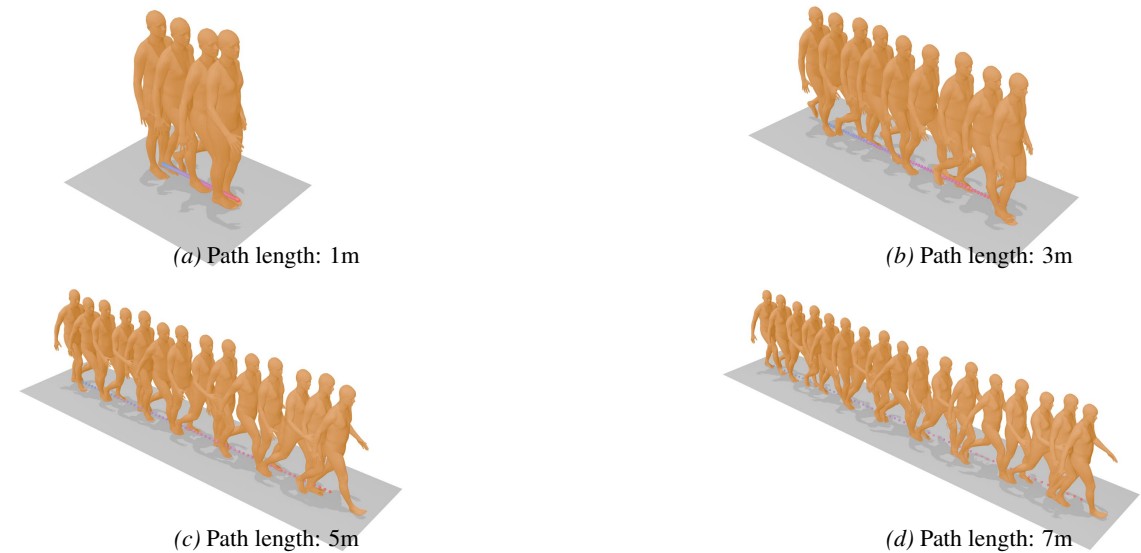

*(a)* Path length: 1m

*(b)* Path length: 3m

*(c)* Path length: 5m

*(d)* Path length: 7m

*Figure 10.* Visualization of generated human motions given the same text description and path direction but different path lengths.

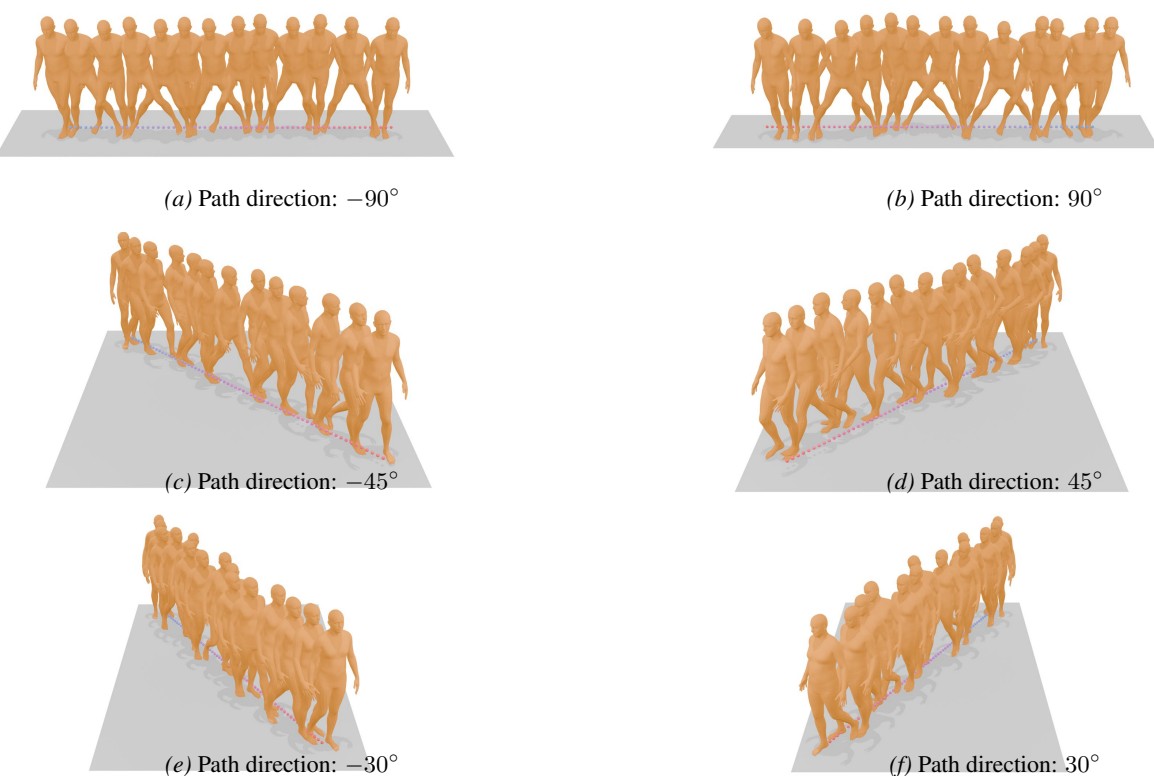

*(a)* Path direction: $-90°$

*(b)* Path direction: $90°$

*(c)* Path direction: $-45°$

*(d)* Path direction: $45°$

*(e)* Path direction: $-30°$

*(f)* Path direction: $30°$

*Figure 11.* Visualization of generated human motions given the same text and path length but different path directions.

Next, we demonstrate the ability to align motion with varying textual descriptions. Using a fixed path, we condition the model on different texts such as *jump*, *run*, *walk as if there are stairs in the front*, and *wave their arms*. As shown in Fig. 12, the generated motions faithfully follow the same path while accurately reflecting the semantics of each instruction.

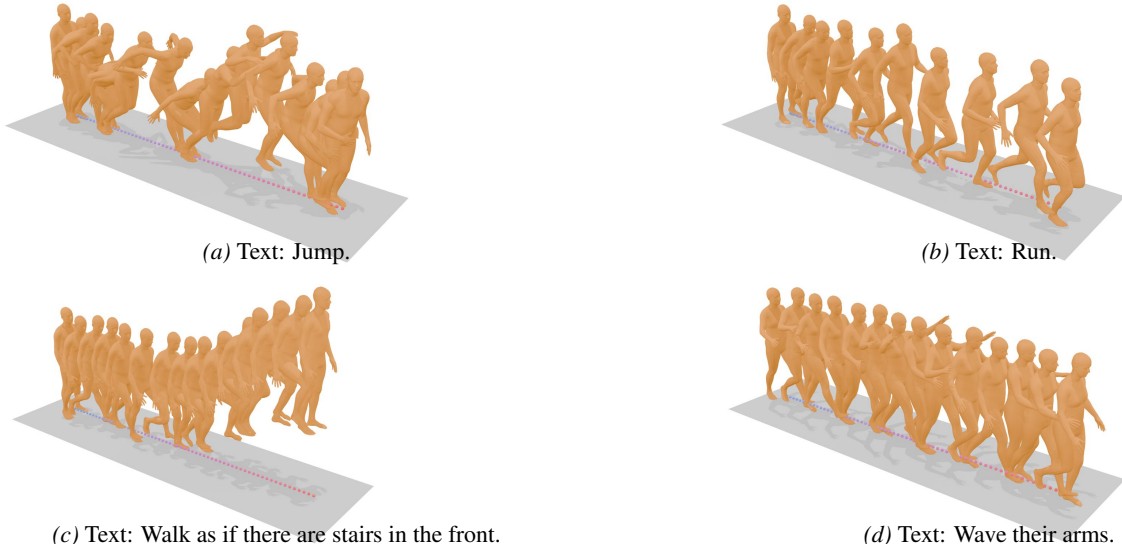

*(a)* Text: Jump.

*(b)* Text: Run.

*(c)* Text: Walk as if there are stairs in the front.

*(d)* Text: Wave their arms.

*Figure 12.* Visualization of generated human motions given the same path direction and length but different text descriptions.

Moreover, we showcase generalization to random combinations of different texts, path lengths and directions in Fig. 13.

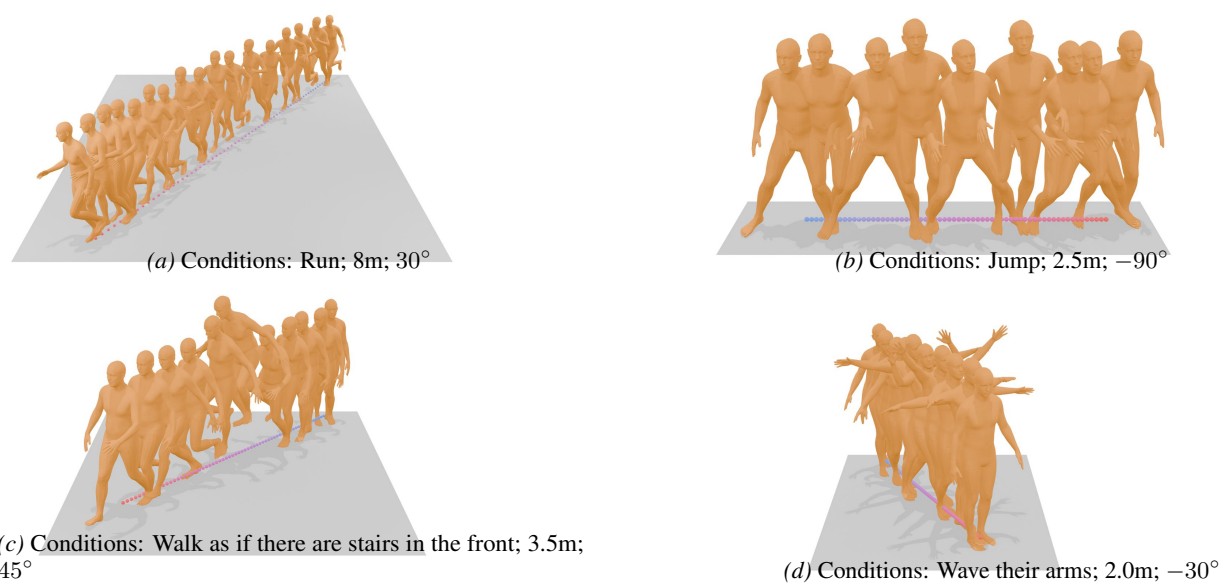

*(a)* Conditions: Run; 8m; 30°

*(b)* Conditions: Jump; 2.5m; −90°

*(c)* Conditions: Walk as if there are stairs in the front; 3.5m; 45°

*(d)* Conditions: Wave their arms; 2.0m; −30°

*Figure 13.* Visualization of generated human motions under varying texts, path lengths, and path directions.

Finally, we present qualitative results of our model on the test set of the HumanML3D dataset shown in Fig. 14. These results highlight the alignment between the generated motions and more complex text and path conditions, demonstrating the model's ability to produce coherent and contextually accurate human motion.

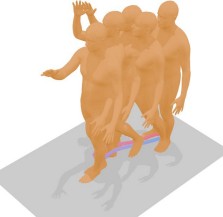

*(a)* Text: The person takes a step and waves his right hand back and forth.

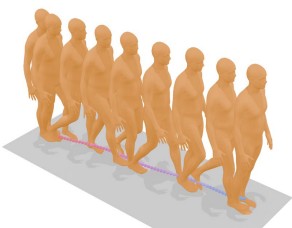

*(b)* Text: A man walks backwards and then stops.

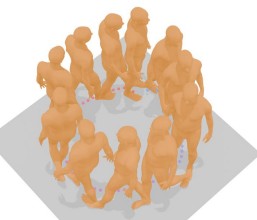

*(c)* Text: A person walks in a circular motion.

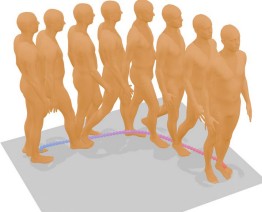

*(d)* Text: A person bends to the right.

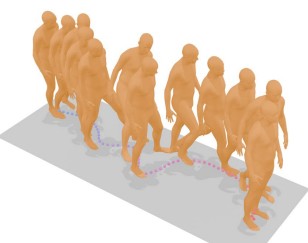

*(e)* Text: A person begins walking forward first with their left foot, taking wide awkward steps as if they are stepping around or over something; begins walking towards the right and then slowly continues to walk to the left, then continues to walk towards the right coming to a stop off to the right side.

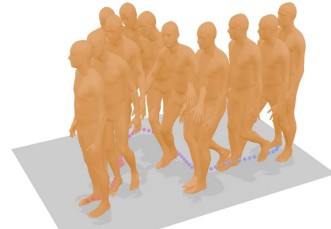

*(f)* Text: The person was pushed but did not fall.

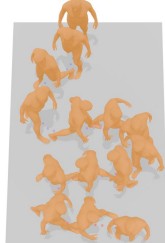

*(g)* Text: A figure tip toes around while walking in a slalom like motion.

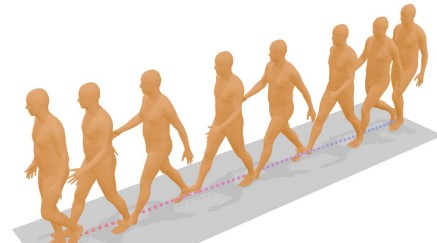

*(h)* Text: A person who is walking moves forward taking six confident strides.

*Figure 14.* Qualitative results from the HumanML3D test set. Text conditions are shown in the subcaptions, and path conditions are visualized as points within the scene.

### A.1.4. ABLATION STUDY ON ADDRESSING PATH OVERFITTING

Apart from the path masking strategy discussed, we also investigated alternative approaches and variants. We hypothesize that the primary source of overfitting is the imbalance between inputs: the path is represented by 64 tokens, whereas the text condition is compressed into a single token by CLIP. To address this, we evaluated mean and max pooling to compress all path features into a single token, following the approach of Reimers & Gurevych (2019), but observed a decline in performance. To preserve both model effectiveness and simplicity, we therefore retain all path tokens and let the transformer learn attention over them. Since we ultimately retain all path tokens, we further explored independent masking (IM), which masks each token independently without segment-level masking, and input perturbation as regularization during

development. We report additional ablation results on pooling, masking, and perturbation, with varied masking rates and noise levels, based on our current model.

*Table 6.* Ablation study on addressing path overfitting.

| Model | R-Prec.↑ | FID↓ | Mean Path Err↓ | Mean Ending Err↓ | Path Err > 60cm↓ | Ending Err > 60cm↓ |
|---|---|---|---|---|---|---|
| Mean Pooling | 0.749 | 0.477 | 0.201 | 0.391 | 0.081 | 0.176 |
| Max Pooling | 0.707 | 0.395 | 0.214 | 0.362 | 0.088 | 0.157 |
| IM 10% | 0.670 | 0.658 | 0.177 | 0.301 | 0.056 | 0.107 |
| IM 50% | 0.728 | 0.298 | 0.156 | 0.284 | 0.037 | 0.099 |
| IM 90% | 0.744 | 0.283 | 0.203 | 0.389 | 0.082 | 0.173 |
| Perturbation 10% | 0.662 | 0.651 | 0.188 | 0.311 | 0.061 | 0.105 |
| Perturbation 50% | 0.671 | 0.501 | 0.173 | 0.277 | 0.051 | 0.092 |
| Perturbation 90% | 0.695 | 0.448 | 0.165 | 0.273 | 0.042 | 0.094 |
| Ours | 0.755 | 0.238 | 0.151 | 0.287 | 0.045 | 0.111 |

As shown above, our full method consistently outperforms variants with mean or max pooling in both text alignment and path-following accuracy. Compared to independent masking and input perturbation to mitigate overfitting, our method achieves a significant improvement in text alignment while maintaining strong path-following performance, striking a better balance between the two objectives. It also achieves a lower FID, indicating higher motion quality.

### A.1.5. TRAINING AND INFERENCE TIME

Lastly, we report and compare our method with baselines for both the training and inference time. We summarize the training time required in the Table 7 below:

*Table 7.* Comparison of training time (hours).

| Ours | T2M-GPT | MDM | OmniControl | MotionLCM |
|---|---|---|---|---|
| 27.1 | 19.7 | 20.9 | 47.1 | 23.2 |

In practice, we adopt the VQ-VAE checkpoint provided by Zhang et al. (2023b) for a fair comparison, and thus exclude its training time (7.1 hours) from the table. For OmniControl and MotionLCM, we also do not include the training time required by pretrained models, as we directly use the released checkpoints. While our method requires longer training time than T2M-GPT due to the longer tokens, it is significantly more efficient than OmniControl, which achieves the closest performance.

*Table 8.* Comparison of inference time (s). For autoregressive models, we report the time required to generate motion sequences of lengths 64 (first) and 196 (second).

| Ours | T2M-GPT | MDM | OmniControl | MotionLCM |
|---|---|---|---|---|
| (0.16, 0.47) | (0.09, 0.27) | 7.43 | 117.26 | 0.05 |

We report the inference time required to generate a sequence of human motion in Table 8. Since the inference time for our method and T2M-GPT depends on the motion sequence length, we provide results for sequences of length 64 and 196. Experiments show that both autoregressive methods are significantly faster than diffusion-based MDM and OmniControl. MotionLCM achieves faster inference through its one-step latent consistency model, but this comes at the cost of weaker performance compared to our method. While our method is slightly slower than T2M-GPT due to the need to compute spatial states on the fly, it delivers substantially better performance.

## A.2. 4D World Generation

### A.2.1. QUALITATIVE RESULTS

While WAVEVERSE can effortlessly generate shorter motions in open or less constrained spaces, we emphasize its ability to handle more challenging scenarios, producing long, semantically and spatially coherent motions within visually complex and spatially constrained environments. Figure 15 showcases qualitative results in such cases, including narrow hallways, intricate layouts, and long human motions. The generated motions align with the surrounding layout, navigating obstacles and fitting within the scene's geometry. Interestingly, the motions sometimes appear to interact with the scene, even though no explicit interaction is modeled. Text prompts are provided in the sub-captions, and we refer to the webpage for corresponding video visualizations.

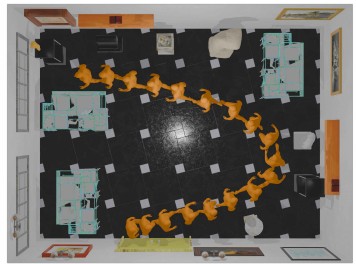

*(a)* A broad gallery; Slowly tour around

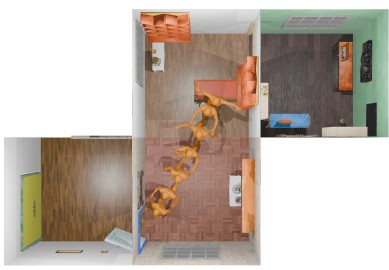

*(b)* A hallway; Wave the arm

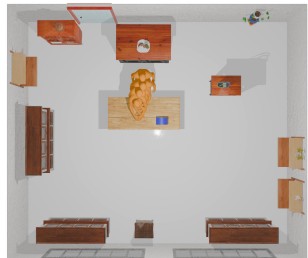

*(c)* A zigzag hallway; Navigate

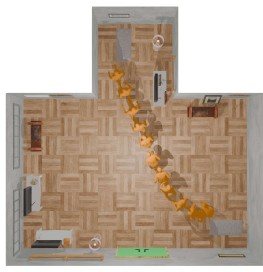

*(d)* A keyhole-shaped hallway; Bend to pick something up

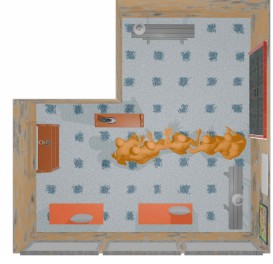

*(e)* A cozy cabin kitchen; Walk to retrieve items

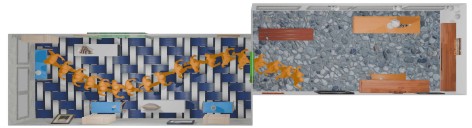

*(f)* A winding corridor; Walk

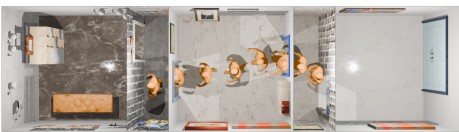

*(g)* A L shape hallway; Quickly Move

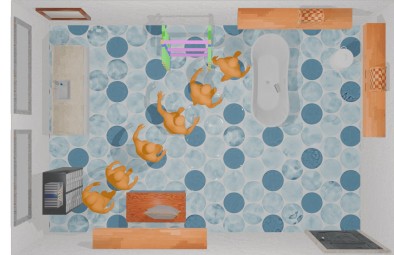

*(h)* A chic bathroom; Walk and almost slip

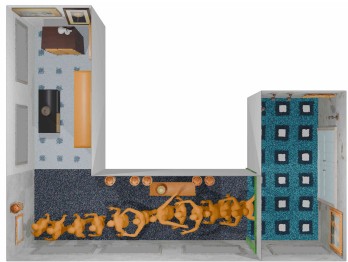

*(i)* A U-shaped hallway; Jump

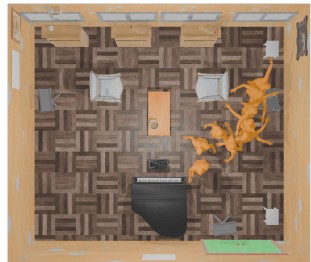

*(j)* A classic music room; Dance

*Figure 15.* Visualization of the generated 4D world, with the environment and motion-generation prompts shown in the subcaptions.

A.2.2. EXECUTION TIME

WAVEVERSE is fully automatic and does not require human interaction, and its components can be parallelized across scenes and radar positions to improve throughput. We report the execution time of each WAVEVERSE module (Fig. 2), measured on a desktop equipped with an RTX 3090 GPU and an i9-11900 CPU, averaged over 10 runs.

- *Input Prompt*. Generating scene and human-shape descriptions takes 1.37 s and 0.56 s, respectively, with most of the latency coming from OpenAI API communication rather than local computation.
- *Environment and Human Shape Generation*. Environment generation and human-shape generation take 105.47 s and 5.17 s, respectively. This includes API calls, mesh creation, object selection and placement in Unity, and loading the fine-tuned LLM checkpoint for human shapes. Importantly, this cost is incurred once per environment and can be amortized over many motion sequences and simulated signals.
- *Motion Description and Path Generation*. Generating motion descriptions and planning paths within the environment (again via API + path search) takes on average 7.03 s.
- *Human Motion Generation*. The Human Motion Generation module takes 20.79 s in total, though only 0.48 s comes from motion generation with our state-aware transformer. The dominant cost is SMPL fitting for the human mesh, which can be further optimized with faster implementations in computer vision.
- *Dielectric Property Generation*. Dielectric properties are precomputed, and the time is already included in environment generation.

We report the time for *RF signal simulation*. Generating a raw measurement for a single radar with 3 transmitters and 4 receivers takes 0.86 seconds for 100k cast rays. In our RF imaging case study with 1,200 radar poses, with a custom CUDA kernel, the runtime can be reduced to 8.97 seconds.

A.2.3. PROMPTS

We provide the adopted prompt in Fig. 16 for the generation of motion descriptions and begin/end points. We also provide the prompts for the human shape generation in Fig. 17 and the dielectric property generation and assignment in Fig. 18, respectively.

```
Motion Description and Path Generation Prompt: You are an experienced human motion designer, expert
in drafting realistic daily human motions within a given environment while considering the context of
the environment. Please assist me in drafting descriptions of daily human motions. You need to give a
text description of motion, including the description of the motion itself, the start and the end
positions. The environment is generated from an environment prompt which will be provided. Please
ensure that the motion description is feasible within the given environment, like the action can be
done by a person within the environment and the start and end points are in the environment. Below is
an example of an environment prompt, the details of the generated environment, and examples of human
motion descriptions. Note: Units for the coordinates are in meters.

For example:
Environment prompt:
A living room.
Environment details:
Floor plan:
living room | [(0, 0), (0, 6), (7, 6), (7, 0)]
Wall height: 2.7
Doors:
door|0|exterior|living room | exterior | living room | [(2.08, 6), (4.08, 6)]
Windows:
window|wall|living room|south|3|0|0 | living room | [(5.27, 0), (6.75, 0)]
window|wall|living room|south|3|1|1 | living room | [(2.70, 0), (4.18, 0)]
window|wall|living room|south|3|2|2 | living room | [(0.27, 0), (1.75, 0)]
Floor objects:
sectional_sofa-0 (living room) | living room | [(5.89, 0, 2.84), (7.05, 0.72, 5.55)]
tv_stand-0 (living room) | living room | [(0, 0, 3.34), (0.54, 0.74, 5.06)]
bookshelf-0 (living room) | living room | [(6.51, 0, 0.13), (7.05, 1.92, 1.27)]
armchair-0 (living room) | living room  | [(3.77, 0, 3.44), (4.62, 1.00, 4.96)]
Wall objects:
painting-0 (living room) | living room | [(6.97, 3.87), (7.00, 4.63)]
```

```
wall-mounted_shelf-0 (living room) | living room | [(4.15, 5.56), (4.95, 6.00)]
Small objects:
55 inch tv-0|tv_stand-0 (living room) | living room | [(0.24, 0.93, 3.80)]
coaster-0|side_table-0 (living room) | living room | [(4.49, 0.73, 5.80)]

Here are some guidelines for you to understand the above environment details:
1. The space is represented in a 'X,Y,Z' coordinate system, where Y represents the height.
2. Whenever there are only two numbers for a coordinate, it represents '(X,Z)', omitting height Y.
3. The detailed environment consists of six parts: Floor plan, Doors, Windows, Floor objects, Wall
objects, and Small objects.
4. The floor plan is represented as: room name | four coordinates of four corners.
5. Doors are represented as: door name | room 1 | room 2 | two coordinates of the projected doors on
X-Z plane (line). The room1 and room2 indicate which rooms the door connects.
6. Windows are represented as: window name | room | two coordinates of the projected doors on X-Z
plane (line). The room indicates which room the window is located in.
7. Floor objects are represented as: floor object name | room | two 3D coordinates which compose the
3D bounding box for the object. The room indicates which room the floor object is located in.
8. Wall objects are represented as: wall object name | room | two 2D coordinates which compose the 2D
bounding box for the projected object on X-Z plane. The room indicates which room the wall object is
located in.
9. The object category is included in its name; you can infer size or height from the name if needed.
10. Do not take the Small objects into consideration when designing the motion.

Motion description examples:
A person walks and gets things from the 'sectional_sofa-0 (living room)' to the 'tv_stand-0 (living
room)', from position '(5.30,4.20)' to position '(0.27,2.80)'.
A person waves hands from the middle of the 'living room' to the 'window|wall|living
room|south|3|2|2', from position '(2.03,3.02)' to position '(1.00,0.20)'.

Motion Design Guidelines:
1. The generated motion description is expected to provide the begin point and the end point; they can
be around objects in the scene or spare spaces in the environment.
2. You need to provide the 2D coordinates of these points on the X-Z plane.
3. You should derive the spatial relations among all objects in the environment.
4. You need to consider the space between objects to ensure that the motion (path) is feasible without
moving objects. In general, more open-space motions are preferred.
5. Objects in the scene do not interact with humans.
6. There might be multiple rooms; you can design a motion from one room to another.
7. Infer from the context to generate diverse actions (run, slip, wave, etc.).
8. Follow the example format exactly: include a complete motion description, optionally provide the
start and end position names, and always include the coordinates in the form 'from position (x1,z1) to
position (x2,z2)'.

Now, you need to design actions for the below prompts:
Environment prompt:
{environment_prompt}
Environment details:
{environment_details}

Generate {motion_number} possible motions for the motion description generation, which should be as
diverse as possible. Strictly follow the format provided in the example. Your response should be
direct and without additional text at the beginning or end.
```

*Figure 16.* Prompt for Motion Description and Path Generation.

**Human Shape Generation Prompt:** Infer one plausible human body shape for the scene {environment_prompt} and return exactly one description listing key physical attributes, with no extra text. Example:
''Average height, tall neck, long arms, and broad shoulders.''

*Figure 17.* Prompt for Human Shape Generation.

**Dielectric Property Generation Prompt:** I have a list of object materials from a 3D asset database: {list_of_object_materials} I need your help to group these materials into broader, high-level material categories. These categories will be used to define radio material properties in an RF simulation engine. Please identify and list appropriate high-level material categories (e.g., metal, plastic, wood, glass, etc.). The goal is to organize the materials in a way that helps assign RF properties during simulation. Use your best judgment based on common material characteristics.

Below is a list of radio materials with their corresponding RF response models and parameter values as defined by the ITU-R P.2040-2 recommendation. I've also included the table of parameters (a, b, c, d) used by the recommendation to model relative permittivity ($\epsilon_r$) and conductivity ($\sigma$) as functions of frequency:

$\epsilon_r = af_{\text{GHz}}^b$ and $\sigma = cf_{\text{GHz}}^d$.

All models assume non-ionized, non-magnetic materials ($\mu_r$ = 1).

{table_of_itu_material_models}.

For the following high-level material categories ({list_of_generated_rf_materials}), please:

1. Assign appropriate values for the parameters (a, b, c, d), following the same functional model as the ITU recommendation.

2. Use informed estimation or analogy with similar existing materials in the ITU-R P.2040-2 table.

The objective is to ensure all new materials have an associated RF response model that reflects the real physical responses to the best of your judgment.

**Material Assignment Prompt:** You are tasked with selecting the most appropriate radio material from the following list based on an object description. You are only allowed to select one material name from the provided list of materials below.

Available materials: {list_of_all_rf_materials}.

Guidelines for selection: 1. First identify the most likely primary material of the object based on common manufacturing practices 2. Consider the bulk material that would dominate RF interactions, not surface coatings 3. For composite objects, select the material that makes up the largest volume 4. If multiple materials could apply, choose the one that would most affect RF propagation 5. Always select the closest matching material from the list only, even if it's not an exact match

Output only the selected material name based on provided object description.

The object: {object_descriptions}.

*Figure 18.* Prompt for Dielectric Property Generation and Assignment.

### A.2.4. PATH PLANNING

Given the input start and end points, we first generate a cost map from the scene layout, which is processed with morphological dilation. We then apply the A* algorithm to find a path between the start and end points.

## A.3. RF Simulation

### A.3.1. GALLERY OF PANORAMIC IMAGING RESULTS

We provide more comparisons of panoramic imaging results with and without spatial phase coherence as in § 4.2, showing that our ray tracing generates high-fidelity signals that can be effectively used for downstream RF applications, whereas baseline simulations without phase coherence fail to produce data of sufficient quality. Notably, the ghost reflections in our results indicate that the simulation captures multipath effects, which learning-based methods struggle to reproduce or guarantee.

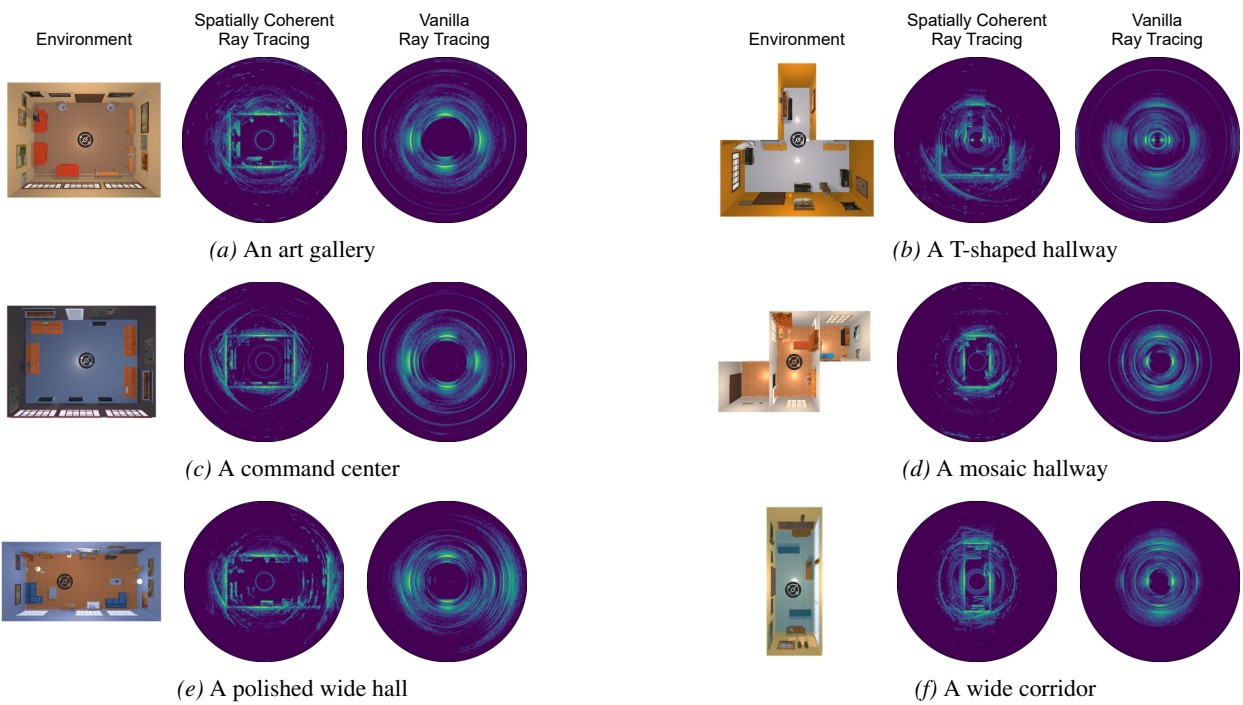

*Figure 19.* Examples of panoramic imaging results. Sensor locations are shown as black icons in environment images.

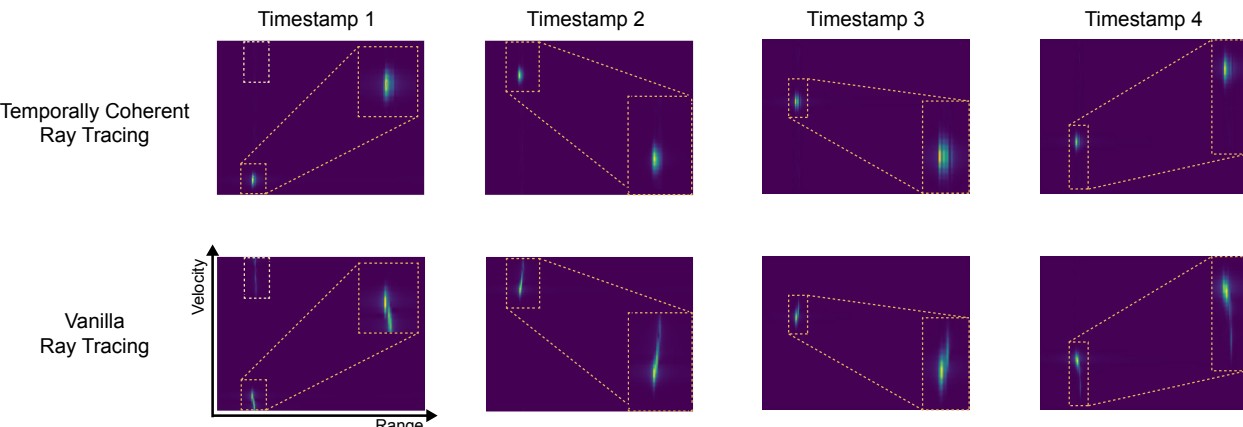

*Figure 20.* Comparison of velocity estimation from Doppler effects between our method and the baseline.

A.3.2. QUALITATIVE RESULTS OF VELOCITY ESTIMATION FROM DOPPLER EFFECTS

We provide qualitative results of velocity estimation from Doppler effects in the video (Doppler_comparison.mp4) attached on our project webpage. In this experiment, we simulate a rigid sphere moving back and forth along a straight line with sinusoidal velocity. A radar is positioned in front of the sphere, and velocity is estimated from Doppler shifts. This task requires precise tracking of phase changes induced by motion across different timestamps. The results are visualized as range–velocity maps at each timestamp, where we expect to observe a sinusoidal velocity pattern over time reflecting the sphere's periodic motion. In addition, a narrow velocity band should appear across several range bins, since the spatial extent of the sphere causes multiple ranges to share the same velocity. The video clearly demonstrates that our method, which preserves temporal phase coherence, produces substantially cleaner range-velocity maps compared to conventional ray tracing. For clarity, Fig. 20 also provides comparisons at four different timestamps, showing that our method outperforms conventional ray tracing.

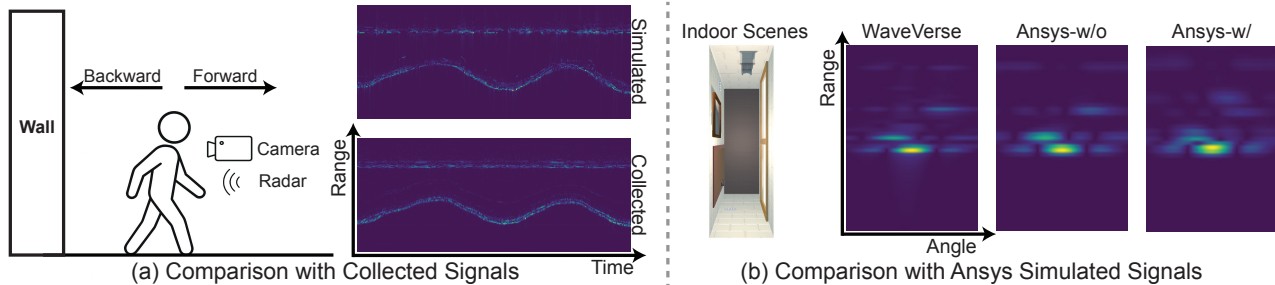

*Figure 21.* Comparison of WAVEVERSE generated RF signals with collected measurements and Ansys HFSS simulations. (a) We compare range–time heatmaps for a subject walking forward and backward in front of a wall, showing close agreement between WAVEVERSE-simulated and radar collected RF signals. (b) We evaluate the range–angle heatmaps simulated by WAVEVERSE against Ansys (Stolarski et al., 2018) HFSS with diffraction and refraction enabled (Ansys-w/) and with these effects disabled (Ansys-w/o).

### A.3.3. COMPARISON WITH REAL MEASUREMENTS AND HFSS SIMULATION

We begin by validating WAVEVERSE against real RF measurements collected using a radar–camera setup, where a subject walks forward and backward in front of a wall. The synchronized camera video is processed with WHAM (Shin et al., 2024) to reconstruct a temporally consistent human mesh sequence. We then rebuild the surrounding environment, including walls, floors, and their spatial layout, and assign material properties based on the surfaces in the scene. Using this reconstructed 4D world, we simulate RF signals with WAVEVERSE and compare the resulting range–time spectrograms. The generated heatmaps, shown in Fig. 21(a), achieve a PSNR of 28.63 dB and a 93.65% similarity in energy distribution, indicating strong alignment with the structural motion patterns and amplitude dynamics of the collected signals, and supporting the correctness and realism of the simulated signals from WAVEVERSE.

We further evaluate WAVEVERSE on additional real measurements covering additional motion patterns and background materials. We collect scenarios involving both back-and-forth walking and unconstrained daily activities in front of four background surfaces, including glass, drywall, brick, and metal. For each sequence, we similarly reconstruct the human motion and surrounding scene geometry, assign material properties according to the observed surfaces, and simulate the corresponding RF measurements with WAVEVERSE. Across these additional real-world settings, WAVEVERSE achieves an average PSNR of 28.39 dB. Moreover, we evaluate multiple signal configurations, including 77–79, 79–81, 77–78, 78–79, 79–80, and 80–81 GHz, using the same motion setup in front of a drywall background. WAVEVERSE achieves an average PSNR of 28.92 dB across these settings, further demonstrating the fidelity of the simulated signals under diverse real-world conditions and frequency configurations.

We also study the effect of material assignment by focusing on three scenes with glass, drywall, and brick backgrounds, where the correctly assigned materials yield an average PSNR of 29.05 dB. Replacing the background material with glass causes only a modest drop to 27.24 dB, while replacing it with metal sharply degrades the PSNR to 19.33 dB. This result shows that direct signal matching is relatively robust to moderate material variations but sensitive to large material mismatches.

To further validate the correctness of the simulated signals, we compare WAVEVERSE with electromagnetic simulations from Ansys (Stolarski et al., 2018) HFSS (High Frequency Structure Simulator), a proprietary EM solver that models wave propagation solving Maxwell's equations. We construct 16 signal-simulation setups from four generated indoor scenes with Ansys-compatible human poses by evaluating each scene under four radar-pose configurations. For every setup, we run HFSS simulations both with and without diffraction and refraction effects enabled. We compare the range–angle spectrograms simulated by WAVEVERSE against both HFSS outputs. When diffraction and refraction are excluded, WAVEVERSE achieves a PSNR of 33.57 dB and 2.12% normalized RMSE. When these effects are included, the results are 31.25 dB and 2.76%, respectively. These findings confirm that WAVEVERSE closely approximates the HFSS-simulated signals with minimal degradation and show that the impact of diffraction and refraction is limited. Moreover, while HFSS requires over one hour per simulation, WAVEVERSE produces comparable results in under one second, offering a scalable alternative.

### A.4. Limitations and Future Work

While WAVEVERSE demonstrates strong performance, limitations remain. First, the current 4D generative pipeline focuses on whole-body dynamics, which is sufficient for most RF sensing tasks but does not yet capture fine-grained interactions such

as typing or manipulating small objects. As a result, the applicability of WaveVerse in interaction-centric scenarios is still limited. However, as world-generation and motion-generation models improve, WaveVerse, as a unified generation-and-simulation framework, can be naturally extended to handle fine-grained human-object interactions. Second, our simulation is built on ray tracing with reflection modeling, which dominates indoor RF propagation and supports most RF sensing tasks. However, more complex phenomena like diffraction around sharp edges and refraction through objects are currently simplified, as in prior work (Cai et al., 2020; Ren et al., 2024). Extending the simulator with UTD-based diffraction and Fresnel-based refraction is a promising direction to reduce this gap, and we leave this for future work. Lastly, while our signal generation pipeline is fully simulation-based, we agree that lightweight, real-data-driven refinement could further enhance fidelity. We consider integrating such refinement into WaveVerse as an interesting direction for future work.

