# OpenReview forum: "Scalable RF Simulation in Generative 4D Worlds"
_ICML.cc/2026/Conference — ICML 2026 regular_

### Official Review · Reviewer_sj6K · 2026-03-11

**Soundness:** 3
**Presentation:** 3
**Significance:** 3
**Originality:** 3
**Overall Recommendation:** 4
**Confidence:** 5

**Summary:**

The paper presents a pipeline for RF simulation in indoor environments. The pipeline automatically constructs 3D scenes with environmental layouts, assigns dielectric properties to objects, and incorporates dynamic human meshes to simulate human presence and motion. RF signals are then generated using a ray-based propagation model. Experiments are conducted on various downstream tasks.

**Compliance With Llm Reviewing Policy:**

Affirmed.

**Key Questions For Authors:**

1.	The dielectric properties are generated using an LLM based on a material library. How do the authors ensure the accuracy and physical validity of these generated parameters? Additionally, how does the method handle objects that are not included in the material library or objects composed of multiple materials?
2.	When generating dynamic human meshes in the scene, how do the authors avoid mesh intersection or penetration with surrounding objects (e.g., walls or furniture)? Could such artifacts affect the RF signal simulation?
3.	The RF simulation appears to rely on a simplified ray propagation model. How does the simulator account for important physical phenomena such as scattering, diffuse reflection, and complex multipath propagation?
4.	Figure 5 shows that the proposed spatial phase coherence method generates panoramic imaging results that better align with wall boundaries and room geometry. However, it is unclear whether this improvement reflects better alignment with true RF signal behavior, or simply better agreement with the geometric layout of the scene. In practice, RF signals often exhibit complex multipath effects that do not necessarily follow visual scene topology. So,how can the authors verify that these results correspond to physically accurate RF signals rather than merely visually consistent geometric reconstructions?
5.	How do the authors evaluate the gap between simulated RF signals and real-world RF measurements? Without quantitative evaluation of this sim2real gap, it is difficult to assess whether models trained on the simulated dataset can generalize to real deployments.
6.	The paper would benefit from a clearer comparison with prior work on RF signal generation, such as “One Snapshot is All You Need: A Generalized Method for mmWave Signal Generation”. That work also addresses RF signal generation and includes more detailed modeling such as indoor multipath reflections, while demonstrating sim-to-real applicability. Could the authors clarify the key differences between the two approaches, particularly in terms of signal modeling fidelity and real-world applicability?

**Limitations:**

yes

**Strengths And Weaknesses:**

Strengths
The paper implements a complete pipeline for RF signal generation, including scene construction, dielectric property assignment, human mesh generation, RF propagation simulation. This framework enables the generation of large-scale RF sensing datasets in diverse simulated environments.

Weaknesses
1.	Reliance on synthetic simulation environments.
2.	Limited reliability of dielectric property generation.
3.	Simplified RF propagation modeling.
4.	Unclear sim-to-real gap.

---

> ### Author Rebuttal · Authors · 2026-03-31
>
> We appreciate the reviewer’s recognition and respond point by point.
>
> 1).**Dielectric property generation and validity**
>
> WaveVerse is a generative framework, so there is no ground-truth material label to recover for each object and no accuracy to measure. The key requirement is therefore not an exact dielectric value, but physical validity of the assigned dielectric properties so that RF propagation remains realistic. To ensure this, in the Sec. 3.1, WaveVerse uses the ITU-R P.2040-2 standard, which provides frequency-dependent parametric models for permittivity and conductivity for 14 common indoor materials. We prompt the LLM to propose additional material categories under the same model, and retain only those whose values fall within valid physical ranges, resulting in a curated library of 24 materials. Thus, the final simulation parameters are physically grounded rather than accepted directly from the LLM. In practice, for each object, the LLM assigns the closest plausible category, and for multi-material objects, it adopts a dominant-material approximation.
>
> 2).**Dynamic human mesh generation**
>
> To reduce human-scene penetration, we reserve clearance from walls and other objects in the path-generation cost map before generating motion trajectories. This makes collision artifacts limited, as shown in the paper. Minor artifacts, such as occasional hand penetration, can occur, but their impact is minimal. Current RF sensing has limited resolution for such fine-scale interactions, and most downstream tasks depend mainly on overall pose and motion rather than precise hand-object interaction, as discussed in the Limitation section.
>
> 3).**Ray tracing details**
>
> Our ray tracing engine is built on Mitsuba 3 and extended for RF simulation. Scattering and diffuse reflection are modeled with a directive scattering pattern that blends Lambertian and specular-lobe components, with Fresnel terms determined by frequency-dependent complex permittivity and conductivity. Complex multipath is captured as in optical path tracing, where each surface interaction recursively samples the next ray.
>
> 4).**Multipath effects**
>
> The simulated signals reflect more than geometric consistency. In Figs. 5 and 18, besides better room-structure alignment, we can also observe arc-shaped structures away from object locations or simple scene contours. These patterns are consistent with multi-bounce propagation and therefore provide qualitative evidence beyond geometry.
>
> We further compare collected and simulated RF signals using latent-space FID from U-Net bottleneck features, which capture RF structure including multipath effects. The resulting FID of 2.88 indicates close agreement with real observations. Together with the comparisons to real measurements and HFSS, this supports that the method captures meaningful RF behavior rather than merely geometrically plausible reconstructions.
>
> 5). **Comparison between the simulated and real signals**
>
> We evaluate this gap in two ways. First, we perform direct signal-level comparisons between simulated and real measurements in controlled settings in Sec. 4.2. Second, we test whether simulated data benefits downstream RF sensing tasks through two case studies. To further strengthen this analysis, in the rebuttal we extend the real-measurement comparison to four background categories, drywall, glass, metal, and brick, and two motion patterns, back-and-forth walking and unconstrained human motion. Across these settings, WaveVerse achieves an average PSNR of 28.39 dB. Combined with the latent-space FID above, these results provide both instance-level and distribution-level evidence that the simulated signals remain close to real measurements.
>
>
> 6).**Comparison with mmGen**
>
> First, mmGen depends on depth-camera capture of real scenes, so its scalability is limited by manual collection and reconstruction quality, whereas WaveVerse generates diverse mesh-based environments directly from prompts. Second, mmGen relies on reconstructed or externally generated human motions, rather than explicitly generating motions conditioned on the scene layout as achieved by WaveVerse. Consequently, mmGen cannot realize the main capability of WaveVerse as a scalable signal simulation framework. Third, mmGen relies on explicit sensor-scene path construction, which becomes expensive as mesh complexity grows, and therefore restricts multipath modeling to a limited subset of two-bounce paths. In contrast, WaveVerse models multi-bounce propagation within a unified phase-coherent framework across all propagation paths in complex scenes. Fourth, mmGen is evaluated only on activity recognition, which makes its effectiveness for RF imaging, especially in large indoor environments with complicated geometry and richer multipath, unclear. In contrast, WaveVerse directly demonstrates effectiveness on other complicated tasks like RF imaging. We will add a clearer comparison with this paper in the Related Work section.

---

> > ### Author Rebuttal · Reviewer_sj6K · 2026-04-05
> >
> > Thank you for the clarifications.

---

### Official Review · Reviewer_KsQh · 2026-03-12

**Soundness:** 3
**Presentation:** 3
**Significance:** 3
**Originality:** 3
**Overall Recommendation:** 4
**Confidence:** 3

**Summary:**

This paper introduces WAVEVERSE, a framework that combines generative 4D world models with physics-based RF simulation to synthesize realistic RF signals at scale. The authors validate their approach against HFSS and real radar measurements, and demonstrate that WAVEVERSE-generated data improves downstream RF imaging and activity recognition tasks.

**Compliance With Llm Reviewing Policy:**

Affirmed.

**Final Justification:**

As stated in my acknowledgement, I maintain my rating.

**Key Questions For Authors:**

1. Please provide an ablation using the same scenes for both simulators to isolate the effect of phase-coherent ray tracing from data diversity. Which component contributes more to the downstream improvements?

2. In Sec. 3.1, the 64 waypoints are described as "spatially evenly spaced." Please specify the exact implementation. Additionally, for the contiguous segment masking in Table 2, what determines segment boundaries, and how are segments selected when multiple unmasked regions remain?

3. In Sec. 3.2, please clarify the maximum bounce depth used in experiments. The paper states that single-bounce paths "typically dominate received energy", can you quantify the energy contribution of higher-order paths in the tested indoor scenes?

4. The comparisons to HFSS and real measurements (Sec. 4.2, Appendix A.3.3) are valuable but the scope is unclear. How many distinct room geometries, human poses, and frequency bands were validated? Do the results generalize to other common RF sensing frequencies?

5. In Table 3 and Fig. 7-8, please confirm that (a) the same network architecture and hyperparameters are used, (b) the same amount of synthetic data is generated for each simulator, and (c) results are averaged over multiple runs for statistical stability.

**Limitations:**

yes

**Strengths And Weaknesses:**

Strengths
1. The paper presents a novel combination of generative 4D world models with physically-grounded RF simulation. The path-conditioned motion generation is a creative design choice that enables scalable data synthesis without manual trajectory specification.

2. The work addresses an important and practical problem: the scarcity of diverse, high-quality RF sensing datasets. The framework has clear potential to benefit the RF sensing community by enabling large-scale data augmentation across different hardware configurations.

3. The experimental evaluation is comprehensive, covering motion generation benchmarks (Table 1-2), phase coherence validation, comparison with HFSS and real measurements, and two downstream case studies.

4. The paper is generally well-written with a clear structure. The problem motivation is well-articulated, and the two-component design (4D world generation + RF simulation) is easy to follow.

Weaknesses
1. Table 3 compares WAVEVERSE with Standard RT, but it is unclear whether both simulators use identical scenes, making it hard to isolate the contribution of phase-coherent ray tracing from data diversity.

2. The validation against HFSS and real-world measurements is limited in scope: real-world comparison uses only one scenario (walking in front of a wall), and HFSS comparison covers 16 configurations at a single frequency band. Generalization to other frequencies remains unclear.

3. The simulator relies primarily on LoS and single-bounce approximation. The paper does not quantify the energy contribution of higher-order bounces or clarify when this approximation may break down.

4. Some algorithmic details lack formal description. The phase-coherent ray tracing (Sec. 3.2) would benefit from pseudocode, and the path masking strategy implementation details are not fully specified.

5. The scalability claim in the title is not well-supported in the main paper.

---

> ### Author Rebuttal · Authors · 2026-03-31
>
> We thank the reviewer for the insightful comments and constructive suggestions. We are particularly grateful for the reviewer’s recognition of the novelty of our framework and its potential value to the RF sensing community. Below, we provide point-by-point responses to the reviewer’s questions.
>
> 1). **Same-scene ablation and the source of downstream improvements**
>
> In our comparisons with baseline simulators, we have already kept the environment layout, human motion, material properties, sensor poses, and all other relevant settings identical across simulators. The only varying factor is the simulator itself. Therefore, the result has already isolated the effect of data diversity and the improvements observed is attributed to the improved fidelity of our simulator.
>
>
> 2). **Implementation of the 64 spatially even waypoints and contiguous segment masking**
>
> The 64 waypoints are computed by reparameterizing the root joint trajectory on the ground plane by arc length. Specifically, we first extract the 2D root position at every frame, then compute the cumulative traveled distance along the trajectory by summing consecutive Euclidean step lengths. We next define 64 target distances that are uniformly spaced between zero and the total path length, and linearly interpolate the 2D position at each target distance. As a result, the extracted waypoints are evenly spaced in terms of physical distance traveled along the path, rather than frame index or elapsed time, thereby avoiding the inclusion of motion velocity or other temporal information in the path condition.
>
>
> About the masking, for each training sample we first determine the target number of masked path tokens by uniformly sampling a mask rate from the predefined range and multiplying it by the path-token sequence length. We then iteratively mask the path tokens in contiguous segments. At each iteration, we identify the currently unmasked regions and select one consecutive unmasked segment of the desired segment length. When multiple candidate segments are available, one is chosen randomly. If the target number of masked tokens has not yet been reached but no unmasked region is long enough to support another full segment, we then randomly mask the remaining available path tokens until the target number is met.
>
>
> 3). **Maximum bounce depth and the contribution of higher-order paths**
>
> We would like to first clarify that, when we state single-bounce paths typically dominate the received energy, higher-order reflections are still included in the ray tracing for all objects, including moving objects. The simplification is only that the hit points from higher-order interactions are not expanded in remapping, which reduces computation because their energy contribution is relatively small.
>
> To answer the question, in the experiments, we set the maximum bounce depth to 5. Across 100 tests over 10 indoor scenes, we find that higher-order paths contribute 5.36% of the total received energy on average. The contribution from higher-order paths involving the human body is even smaller. We will clarify these details in the revision to avoid confusion.
>
>
> 4). **Validation against HFSS and real measurements**
>
> For the HFSS comparison, we conduct experiments in the 77–81 GHz band across four distinct room geometries and four different poses. For the real-measurement comparison, due to hardware availability, the current draft uses the 77–81 GHz band. To strengthen the scope, we additionally evaluate a 64–68 GHz setting in comparison with HFSS, achieving a PSNR of 32.82 dB. For the real-measurement validation, we additionally collect 8 trajectories where a person moves back and forth or performs unconstrained motion in front of different backgrounds, including glass, drywall, brick walls, and metal doors, yielding an average PSNR of 28.39 dB. We also test six frequency-band settings in front of drywall, including 77–79, 79–81, 77–78, 78–79, 79–80, and 80–81GHz, obtaining an average PSNR of 28.92 dB with our available hardware. These results support the strong generalization ability of the method.
>
>
> 5). **Details in comparison**
>
> Yes, we use the same network architecture and hyperparameters, generate the same amount of synthetic data for each simulator, and report the results averaged over three runs with the original evaluation protocol.

---

> > ### Author Rebuttal · Reviewer_KsQh · 2026-04-04
> >
> > Thank you for your detailed and thoughtful reply. I will maintain my rating accordingly.

---

> > > ### Author Response · Authors · 2026-04-04
> > >
> > > We appreciate the reviewer’s careful consideration of our response. It is encouraging to know that our clarifications and the additional experiments helped address the questions raised. Thank you again for your constructive comments and time.

---

### Official Review · Reviewer_d6U3 · 2026-03-13

**Soundness:** 4
**Presentation:** 4
**Significance:** 3
**Originality:** 4
**Overall Recommendation:** 5
**Confidence:** 4

**Summary:**

This paper proposes WAVEVERSE, which integrates generation and physical simulation, to address the challenge of insufficient large-scale and diverse RF (Radio Frequency) sensing datasets (including data collection costs due to diverse indoor layouts, activities, and individual differences, as well as difficulties in data reuse due to hardware differences such as bandwidth and antenna placement).
WAVEVERSE combines (i) 4D world generation using LLMs (3D indoor environments + time-varying human motion) with (ii) physics-based RF signal simulation (ray tracing) to generate phase-coherent signals across antennas and time.
Evaluation demonstrates: (1) text-plus-path-conditioned motion generation performance on HumanML3D, (2) success rate and diversity of 4D scene generation, (3) phase-dependent task performance with phase-coherent ray tracing (panoramic imaging, respiration estimation, Doppler velocity estimation), (4) signal consistency with real measurements and Ansys HFSS, and (5) utility demonstrated through case studies based on public data (RF imaging depth estimation, behavior recognition).

**Compliance With Llm Reviewing Policy:**

Affirmed.

**Final Justification:**

The rebuttal from the authors has addressed my concerns, keeping the score.

**Key Questions For Authors:**

- Regarding the reproducibility of LLM usage, could you specify the LLMs used for scene generation, action description generation, dielectric property generation/assignment, and body shape generation (including inference settings such as model names and prompt consistency)? This would alter the evaluation of reproducibility and result robustness.
- The paper's primary motivation is that ``RF data is difficult to reuse due to differences in bandwidth, antenna configuration, modulation schemes, etc.'' However, the experiments primarily focus on imaging/HAR/fidelity comparisons under limited settings, and it does not appear to directly verify whether the approach is truly effective across different sensor configurations. Is it possible to directly demonstrate WAVEVERSE's claimed flexible sensor configuration and generalization to unseen conditions across multiple settings with significantly altered frequency bands, bandwidths, antenna configurations, scanning methods, etc.? Demonstrating this would significantly enhance the paper's significance.

**Limitations:**

Yes

**Strengths And Weaknesses:**

Strengths:
- The methodology is structured with clear diagrams and section organization to facilitate understanding: separation of 4D world generation and RF simulation, path conditioning, and phase matching problem formulation.
- The appendix includes prompt examples (action descriptions, start/end points, body shape generation, dielectric property generation/assignment) and path planning, demonstrating concrete implementation details.
- Quantitative breakdowns of execution time (dominated by OpenAI API latency, Unity mesh generation, SMPL fitting, etc.) provide supporting evidence for scalability claims.

Weaknesses:
- Spatial phase matching is designed to map the fixed path set obtained by the representative radar to each radar, discarding obstructed paths. The procedure for generating newly emerging paths specific to each radar is difficult to discern from the text. The quantitative analysis of how much this approximation affects specific configurations (such as large attitude differences or numerous obstructions) appears limited in the text.
- The collision ratio of 2.35% and cumulative collision depth of 12.23 cm reported as metrics for 4D scene generation suggest average environmental consistency. However, depending on the application, the impact of this level of penetration on learning remains a concern.
- Multiple instances of LLM usage are observed (scenes/actions/materials, etc.), but at least in the appendix sections provided, while the existence of OpenAI API communication is mentioned, details such as the model name and temperature used for inference are not explicitly stated (potentially lacking from a reproducibility perspective).

---

> ### Author Rebuttal · Authors · 2026-03-31
>
> We sincerely thank the reviewer for the time spent on our work and the insightful and thoughtful assessment and suggestions for our work, and are especially encouraged by the positive assessment.  Below, we address the questions raised by the reviewer and provide corresponding clarifications.
>
>
> 1). **Details about the used LLM**
>
> For all LLM-based components, including scene generation, action description generation, body shape generation, and dielectric property generation and assignment, we use GPT-4o through the OpenAI API with the LangChain interface. To ensure consistency across runs, we use fixed prompts, which are already included in the Appendix. As noted in the paper, we will release the codebase upon acceptance, which will include the prompt templates and interfaces needed to reproduce more easily.
>
> 2). **Generalization across sensor configurations**
>
> WaveVerse is designed to support scalable RF simulation across diverse hardware and sensing setups, motivated by the practical challenge that RF data is often difficult to reuse across systems with different configurations. We would like to clarify, however, that our objective is to enable high-fidelity signal synthesis for diverse sensor configurations, rather than to make a model trained under one configuration directly generalize to another configuration. The latter is a different problem setting and is beyond the scope of this paper.
>
> Our paper has provided strong empirical evidence that WaveVerse remains effective across varied sensing setups and diverse RF tasks. At the method level, WaveVerse is built on an explicit CIR-based physical simulator with our proposed phase-coherent ray tracing, rather than relying on a learned sensor-specific generator. This design naturally supports changes in hardware setup through the same physical formulation. At the experimental level, we validate WaveVerse along multiple complementary axes, including both rotating (synthetic aperture) and static scanning patterns, both real-world measurements and Ansys HFSS for fidelity, and multiple phase-sensitive benchmarks where spatial and temporal phase coherence is essential. Finally, we demonstrate end-to-end utility in two downstream case studies, RF imaging and human activity recognition, where the benefits continue to grow as more simulated data is added. Collectively, these experiments demonstrate the strong generalizability of WaveVerse across diverse configurations, including change in frequency bands, antenna configurations, and scanning methods.
>
> We also extend the RF fidelity evaluation in the rebuttal to a broader set of frequency-band configurations, further showing that WaveVerse generalizes across different frequency-related RF configurations more explicitly. Specifically, we compare collected real measurements and simulated signals under 77–79 GHz, 79–81 GHz, 77–78 GHz, 78–79 GHz, 79–80 GHz, and 80–81 GHz with our available hardware in two scenarios, back-and-forth motion along a line and unconstrained movement in front of the sensor. Across these settings, WaveVerse achieves an average PSNR of 28.92 dB. We further evaluate a 64–68 GHz configuration through comparison with Ansys HFSS, where WaveVerse achieves a PSNR of 32.82 dB. Together, these results provide additional evidence that WaveVerse generalizes well across different RF frequencies and frequency-band configurations.
>
> Beyond this, we agree that enabling a downstream model trained under one sensor configuration to remain effective under different sensor configurations is an interesting direction. This would require a systematic study of model-level transfer across changes in different configurations, which goes beyond the scope of the current paper. We view this as a natural next step and a promising direction for follow-up work.

---

> > ### Author Rebuttal · Reviewer_d6U3 · 2026-04-04
> >
> > Thank you for the helpful clarification. My concern about the reproducibility of the LLM components is largely resolved, since the rebuttal now specifies the use of GPT-4o via the OpenAI API/LangChain interface and fixed prompts. My concern about generalization across sensor configurations is partially resolved: the rebuttal clarifies that the intended claim is high-fidelity signal synthesis under diverse sensor configurations, rather than downstream model transfer across configurations, and the added multi-band fidelity results strengthen the paper.

---

> > > ### Author Response · Authors · 2026-04-04
> > >
> > > We sincerely thank the reviewer for considering our response. We are grateful that our clarification and the experiments have addressed your concerns. Thank you again for your time and valuable feedbacks.

---

### Official Review · Reviewer_FqHQ · 2026-03-13

**Soundness:** 3
**Presentation:** 3
**Significance:** 3
**Originality:** 2
**Overall Recommendation:** 4
**Confidence:** 4

**Summary:**

The paper proposes WaveVerse, a prompt-driven framework that generates realistic RF signals in dynamic 4D environments using a language-guided world generator and a physics-based simulator, producing high-fidelity signals. It has been demonstrated WaveVerse generated data can improve the downstream tasks.

**Compliance With Llm Reviewing Policy:**

Affirmed.

**Final Justification:**

The rebuttal addressed most of my main concerns and I keep my prior rate.

**Key Questions For Authors:**

1. How sensitive are the downstream RF results to errors in LLM-assigned dielectric materials and other generated intermediate attributes, and can the authors quantify this with perturbation or ablation studies?

2.The paper validates against real measurements in one walking setup and against HFSS on 16 static scenes; how well do the fidelity claims hold across more diverse real environments, object materials, and motion patterns?

3.Since diffraction and refraction are simplified, in what classes of indoor scenes do the authors expect the current simulator to fail materially, and how should practitioners detect those regimes?

4.Several baselines are adapted from trajectory-conditioned or otherwise different settings; can the authors better disentangle gains from the proposed modules versus gains from task formulation or adaptation mismatch?

5.Can the authors clarify the intended deployment scope across radar hardware, including what aspects have been tested empirically versus argued from the CIR formulation?

**Limitations:**

The authors have not discussed the limitations. The current proposed methods mainly target RF signal for indoor scenes with whole human motions. Would be nice to explore the feasibility in other scenarios. Meanwhile, the method relies on simulator data and there might errors between simulation and real situation.

**Strengths And Weaknesses:**

1. The technical ideas are plausible and supported by experiments, but the evidence is not yet strong enough to remove concerns about simulator assumptions, intermediate LLM-generated components, and the breadth of real-world validation.
2. The paper is clearly written and well structured and easy to read. The paper discussed the realted literature and clearly explains how it differs from prior work.
3. The paper studies an important problem and shows promising empirical gains, but the demonstrated impact is still limited by narrow task coverage, modest real-world validation, and simulator assumptions that may restrict generalization beyond the tested setups.
4. The paper combines several existing ingredients in a thoughtful way and adds meaningful engineering around path-conditioned motion generation and phase-coherent ray tracing, but the overall contribution feels more incremental and integrative than fundamentally novel.

---

> ### Author Rebuttal · Authors · 2026-03-31
>
> We sincerely appreciate the constructive suggestions. Below we address the questions.
>
> 1). **Sensitivity to materials and other attributes**
>
> We first clarify that different generated attributes affect the pipeline differently. Attributes such as scene layout, human shape, and human motion mainly change the geometry and dynamics of the generated scene itself. Perturbing them therefore changes the simulated scene. By contrast, dielectric assignment directly affects RF propagation under the same scene. We focus our analysis on dielectric materials as the most direct response.
>
> WaveVerse is a generative framework, so there is no ground-truth material label to recover for each object. The key requirement is therefore not an exact dielectric value, but physically valid dielectric properties for RF simulation with reasonable semantic alignment. In our design, these properties are not arbitrary LLM outputs. The LLM assigns only high-level material categories, which are mapped to a curated library of physically grounded dielectric parameters.
>
> With this design in place, we quantify sensitivity to dielectric assignment through deliberate perturbations and find task-dependent effects. For direct signal-fidelity evaluation, we collect real RF measurements of a person moving in front of brick and drywall backgrounds, and simulate each scene under the correct material assignment as well as glass and metal substitutions. The correct assignment yields 29.05 dB PSNR. Replacing it with glass causes only a modest drop to 27.24 dB (-1.81 dB), whereas replacing it with metal sharply degrades PSNR to 19.33 dB (-9.72 dB), showing that direct signal matching is sensitive to large material mismatch. By contrast, for the network-based RF imaging task, sweeping wall permittivity over [3, 50] in 5 scenes yields depth-prediction PSNR values between 31.70 dB and 32.79 dB, indicating robustness to dielectric perturbations from network generalization.
>
> 2). **Fidelity across different real scenes**
>
> WaveVerse shows strong fidelity under broader real-world conditions. We collect additional real measurements with glass, brick, drywall, and metal backgrounds, and consider back-and-forth walking and unconstrained daily activities. Across eight additional trajectories, WaveVerse achieves a PSNR of 28.39 dB. We further test multiple signal configurations, 77–79, 79–81, 77–78, 78–79, 79–80, and 80–81 GHz, and obtain an average PSNR of 28.92 dB. These results support our fidelity claims across broader environments, materials, motions, and sensing configurations.
>
> 3). **Discussion about diffraction and refraction**
>
> WaveVerse remains accurate in reflection-dominated indoor scenes, and may degrade when diffraction or refraction becomes a major contributor to the received signal, as discussed in the Limitations. For the indoor scenes studied in our paper, these effects are limited, as the HFSS comparison in the paper and supplementary shows only a modest PSNR drop when diffraction and refraction are included, from 33.57 dB to 31.25 dB. WaveVerse is well suited to typical indoor layouts, and is less suited to cases such as edge-dominated cluttered settings. In practice, these regimes can be detected by persistent unexplained peaks or abnormal range-angle heatmap artifacts.
>
> 4). **Gains from proposed modules versus task formulation**
>
> The primary gains come from the proposed modules, as directly supported by Table 2. Specifically, Table 2 isolates our two modules on top of T2M-GPT and shows that both path masking and the state-aware transformer yield substantial improvements, while the original T2M-GPT baseline underperforms other methods. For the diffusion baselines, we adapt them as fairly as possible, test multiple variants, and report the best-performing one for each, as shown in Appendix A.1.1 and A.1.2. Lastly, autoregressive methods also have an inherent advantage because they do not require pre-specifying motion length.
>
> 5). **Deployment scope across radar hardware**
>
> WaveVerse is intended for radar hardware governed by the CIR-based signal model. Together with the rebuttal experiments above, our paper empirically validates this scope under different hardware configurations, including changes in frequency band, sampling rate, scanning method, and antenna layout. These experiments cover the main sensing factors that directly affect the CIR in our setting and provide strong evidence that WaveVerse transfers across representative radar configurations. Beyond the tested settings, we expect the simulator to generalize to other specific hardware parameters governed by the same signal model, since it relies on an explicit CIR-based formulation and phase-coherent ray tracing rather than learned sensor-specific assumptions.
>
> 6). **Limitation Section**
>
> We discuss limitations at the end of the appendix due to page limits. In the revision, we will point readers more clearly to it in the main text and expand it to include the reviewer’s suggestions.

---

### Decision · Program_Chairs · 2026-04-30

**Decision:**

Accept (regular)

**Comment:**

The paper proposes a framework for synthesizing realistic RF signals at scale by combining a language-guided world generator with a physics-based simulator. The reviewers raised several concerns, including sensitivity to input attributes, generalization across diverse real-world scenes and sensor configurations, validation against HFSS and real measurements, and comparisons between simulated and real signals. The rebuttal has addressed most of these issues well. After the discussion stage, all reviewers are positive about the paper. They generally appreciate the plausible idea, the importance of the problem, the completeness of the pipeline, and the comprehensiveness of the experiments.